# PD-L1[+] plasma cells suppress T lymphocyte responses in patients with sepsis and mouse sepsis models

Morgane Gossez[1,2], Clara Vigneron[2], Alexandra Vandermoeten[3], Margot Lepage [1,2], Louise Courcol[2], Remy Coudereau[1,4], Helena Paidassai [2], Laurent Jallades[2,5], Jonathan Lopez[6], Khalil Kandara[1], Marine Ortillon[1], Marine Mommert[4], Astrid Fabri[1], Estelle Peronnet[4], Clémence Grosjean[4], Marielle Buisson[7], Anne-Claire Lukaszewicz[4,8], Thomas Rimmelé[4,8], Laurent Argaud [9], Martin Cour [9], REALISM study group*, RICO study group*, Bénédicte F. Py [2], Olivier Thaunat [2], Thierry Defrance [2,19], Guillaume Monneret [1,4,19] & Fabienne Venet [1,2]✉

Sepsis, a leading cause of death in intensive care units, is associated with immune alterations that increase the patients' risk of secondary infections and mortality, so better understandings of the pathophysiology of sepsis-induced immuno-suppression is essential for the development of therapeutic strategies. In a murine model of sepsis that recapitulates immune alterations observed in patients, here we demonstrate that PD-L1[+]CD44[+]B220[Low]CD138[+]IgM[+] regulatory plasma cells are induced in spleen and regulate ex vivo proliferation and IFNɣ secretion induced by stimulation of T splenocytes. This effect is mediated both by cell-cell contact through increased PD-L1 expression on plasma cells and by production of a soluble factor. These observations are recapitulated in three cohorts of critically ill patients with bacterial and viral sepsis in association with increased mortality. Our findings thus reveal the function of regulatory plasma cells in the pathophysiology of sepsis-induced immune alterations, and present a potential therapeutic target for improving immune cell function impaired by sepsis.

Sepsis, defined as a life-threatening organ dysfunction caused by a dysregulated host response to infection[1], is a highly frequent syndrome affecting around 50 million people worldwide and leading to 11 million deaths annually[2]. Recently, the severe cases of SARS-CoV-2 infection pandemic, which corresponds to a viral sepsis, has led to a major influx of patients to the ICU and will most likely increase these numbers. Sepsis thus represents a major healthcare priority worldwide, as recently acknowledged by the WHO[3].

Sepsis initiates a complex immune response that varies over time, with the concomitant occurrence of both pro- and anti-inflammatory

[1]Hospices Civils de Lyon, Immunology Laboratory, Lyon-Sud & Edouard Herriot University Hospitals, Lyon, France. [2]CIRI, Centre International de Recherche en Infectiologie, Univ Lyon, Inserm U1111, Université Claude Bernard-Lyon 1, CNRS, UMR5308, ENS de Lyon, Lyon, France. [3]Service Commun des Animaleries de Rockefeller (SCAR) - Université Claude Bernard lyon1, Structure Fédérative de Recherche (SFR) Santé Lyon Est, Lyon, France. [4]EA 7426 "Pathophysiology of Injury-Induced Immunosuppression" (Université Claude Bernard Lyon 1, Hospices Civils de Lyon, bioMérieux), Lyon, France. [5]Hospices Civils de Lyon, Lyon Sud University Hospital, Hematology Laboratory, Pierre-Bénite, France. [6]Hospices Civils de Lyon, Biochemistry and Molecular Biology department, Lyon Est Faculty of Medicine, Université Claude Bernard Lyon 1, Université de Lyon, Lyon Sud University Hospital, Pierre-Bénite, France. [7]Centre d'Investigation Clinique de Lyon (CIC 1407 Inserm), Hospices Civils de Lyon, Lyon, France. [8]Hospices Civils de Lyon, Anesthesia and Critical Care Medicine Department, Edouard Herriot Hospital, Lyon, France. [9]Hospices Civils de Lyon, Medical Intensive Care Department, Edouard Herriot Hospital, Lyon, France. [19]These authors jointly supervised this work: Thierry Defrance, Guillaume Monneret. *Lists of authors and their affiliations appear at the end of the paper. ✉e-mail: fabienne.venet@chu-lyon.fr

mechanisms. As a result, some patients may rapidly develop signs of profound immunosuppression with altered innate and adaptive immune responses associated with frequent viral reactivations, increased susceptibility to secondary infections (mostly opportunistic) and high mortality[4]. Thus immune-adjuvant treatments targeting sepsis-induced immunosuppression are proposed in the most immunosuppressed patients as a novel therapeutic approach in sepsis[5,6].

Mechanisms leading to sepsis-induced immune alterations are still being explored. Among others, the expansion of immune cell subpopulations with immunoregulatory functions has been described[4]. For example, the increased proportion of CD4+CD25+CD127[low]Foxp3+ regulatory T cells has been well documented both in murine models of sepsis and in patients[7]. Similarly, within the myeloid lineage, the induction of monocytic and granulocytic myeloid-derived suppressor cells (MDSCs) has been repeatedly observed[8].

Cells from the B cell lineage are also affected by sepsis. In particular, decreased circulating B cell numbers and altered effector functions have been described in patients and septic mice[4,9,10]. In murine models of autoimmune diseases, cancer and infections, B cells with regulatory functions (termed Breg) have been described to emerge at different developmental stages from immature B cells to plasma cells (PCs)[11–13]. Only a few studies evaluated the induction of Bregs in murine models of sepsis, and their reported phenotype and mechanisms of action appeared quite heterogeneous[14,15]. Data are even scarcer in the human clinical setting, in particular for severe COVID-19, a condition in which no functional study has evaluated the induction of Bregs so far.

In the current study, we show that regulatory PCs are induced in a murine model of sepsis and in three cohorts of critically ill patients diagnosed with either bacterial sepsis or severe COVID-19. We demonstrate that these induced PCs contribute to sepsis-induced immune alterations by inhibiting T-cell proliferation and IFN-γ secretion ex vivo through increased IL-10 production and elevated PD-L1 expression, independently of regulatory T cells.

## Results

### Murine model of resuscitated CLP recapitulates immune alterations observed in septic patients
We developed a pre-clinical murine model of sepsis by submitting mice to a mild cecal ligation and puncture (CLP). Control group of Sham-treated animals were submitted to laparotomy without ligation nor puncture. Both groups received antibiotics, analgesics and a lactate Ringer solution.

Surgery induced weight loss in both groups, which was significantly more profound and persistent in CLP mice compared with Sham animals (Fig. 1a). CLP was associated with a significant increase over time of severity score, whereas it remained low in Sham animals (Fig. 1b). At 48 h after surgery, no mortality was observed in the group of Sham animals and mortality was low (<10%) in CLP mice (Fig. 1c).

After 48 h, CLP was associated with a significant decrease in the number of total splenocytes (Supplementary Fig. 1A). Absolute counts and percentages of dendritic cells and NK cells but not of other myeloid cells subpopulations were significantly decreased after CLP (Supplementary Fig. 1B–F). Absolute counts (Fig. 1d) but not percentages (Supplementary Fig. 2A) of CD4+ and CD8+ T lymphocytes were simultaneously decreased compared with Sham animals. Among CD4+ T cells, absolute counts of naive and effector T cells tended to decrease after CLP (Supplementary Fig. 2B–D), while percentages of central memory cells increased in CLP mice (Supplementary Fig. 2C). Among CD8+ T cells, absolute count of effector T cells significantly decreased after CLP (Supplementary Fig. 2G). Finally, the percentage of CD4+CD25+CD127[low] regulatory T cells (Treg) significantly increased (Fig. 1e) while their absolute count was maintained in CLP mice (Supplementary Fig. 2H). Accordingly, the proportions of apoptotic and necrotic cells were increased among CD4+ and CD8+ T lymphocytes in CLP mice compared with Sham animals (Fig. 1f). CD4+ and CD8+ T

lymphocytes also presented significantly increased expressions of CD69 and PD-1 in mice after CLP compared with Sham animals (Fig. 1g) suggesting an activated/exhausted phenotype.

This was concordant with functional alterations induced after CLP in remaining T lymphocytes as illustrated by the reduced proliferative response of total T cells and of both CD4+ and CD8+ T lymphocytes and reduced IFNγ secretion after TCR stimulation in CLP mice as compared with Sham animals (Fig. 1h–i, Supplementary Fig. 3A, B). Such altered functionality was also evidenced after non-specific mitogen stimulation (Supplementary Fig. 3C) and after ex vivo antigen-specific re-stimulation of Ag-experienced T cells. Regarding the latter, IFNγ secretion in the supernatant and numbers and percentages of IFNγ producing CD4+ and CD8+ T splenocytes were reduced after CLP upon antigenic re-stimulation ex vivo of splenocytes from immunized CLP mice compared with Sham animals (Fig. 1j, k, Supplementary Fig. 3D).

In total, we designed a pre-clinical murine model of sepsis, which recapitulated immune alterations routinely observed in septic patients[4].

### Regulatory B lymphocytes are not induced in mice after CLP
In this model of sepsis, we next evaluated whether regulatory B cells were induced in the spleen after CLP and participated in sepsis-induced T-cell dysfunctions.

Although a mild but significant decrease in B220+ B splenocyte counts was observed after CLP compared with Sham animals (Fig. 2a), the percentages of total B splenocytes and of CD1d[High]CD5+ regulatory B10 cells[16,17] were not significantly modified (Fig. 2b, c). Expressions of CD40, CD86, PD-1, PD-L1, TIM-3 and LAG-3 were not changed on B220+ lymphocytes after sepsis (Fig. 2d, e). Only major histocompatibility complex class II (MHC class II) expression was significantly decreased on B splenocytes after CLP (Fig. 2d) in accordance with observations made in septic patients[9].

Transcriptomic analysis of purified B220+ B splenocytes showed that, among the 540 mRNA detected in B cells, only 6 (<2%) were significantly differentially expressed between B220+ B splenocytes from Sham and CLP animals (Fig. 2f).

Finally, co-culture experiments between T lymphocytes purified from Sham animals and B220+ B splenocytes purified either from Sham or CLP mice showed that B cells purified after sepsis did not significantly modulate T-cell proliferative response or IFNγ release after TCR stimulation (Fig. 2g, h).

Overall, these results argue against the induction of regulatory B cells in spleen in this model of sepsis.

### Regulatory CD138+IgM+ PCs are induced in spleen after CLP
In accordance with observations made in septic patients[9], our results revealed the induction of a population of B220[low]CD138+ splenocytes with plasma cell (PCs) morphology after CLP (Fig. 3a, b). We confirmed that the percentages and absolute counts of these cells significantly increased in the spleen after CLP as compared with Sham animals (Fig. 3c, d). We showed that these cells expressed high levels of Prdm1 (coding for BLIMP1) and Irf4 mRNAs but low levels of pax5 and bcl6 mRNAs compared with B cells after CLP (Fig. 3e). In addition, intracellular BLIMP1 expression was significantly higher in B220[low]CD138+TACI+ splenocytes compared with B cells from same CLP animals (Fig. 3f).

The majority of these sepsis-induced PCs expressed membrane and intracellular IgM (Fig. 3g) and were characterized by higher density of expression of B220 and MHC class II and reduced expression of CD138 as compared to Sham animals (Fig. 3h).

Transcriptomic analysis of PCs purified from CLP mice illustrated differences in gene expressions related to differentiation profile (i.e. decreased Cd28, Ms4a1 coding for CD20, Slamf7, Irf4, Prdm1 coding for BLIMP1 but increased Xbp1 mRNA levels) and BCR signaling (i.e. decreased Cd19 and Cd22 mRNA levels, compared with PCs from Sham

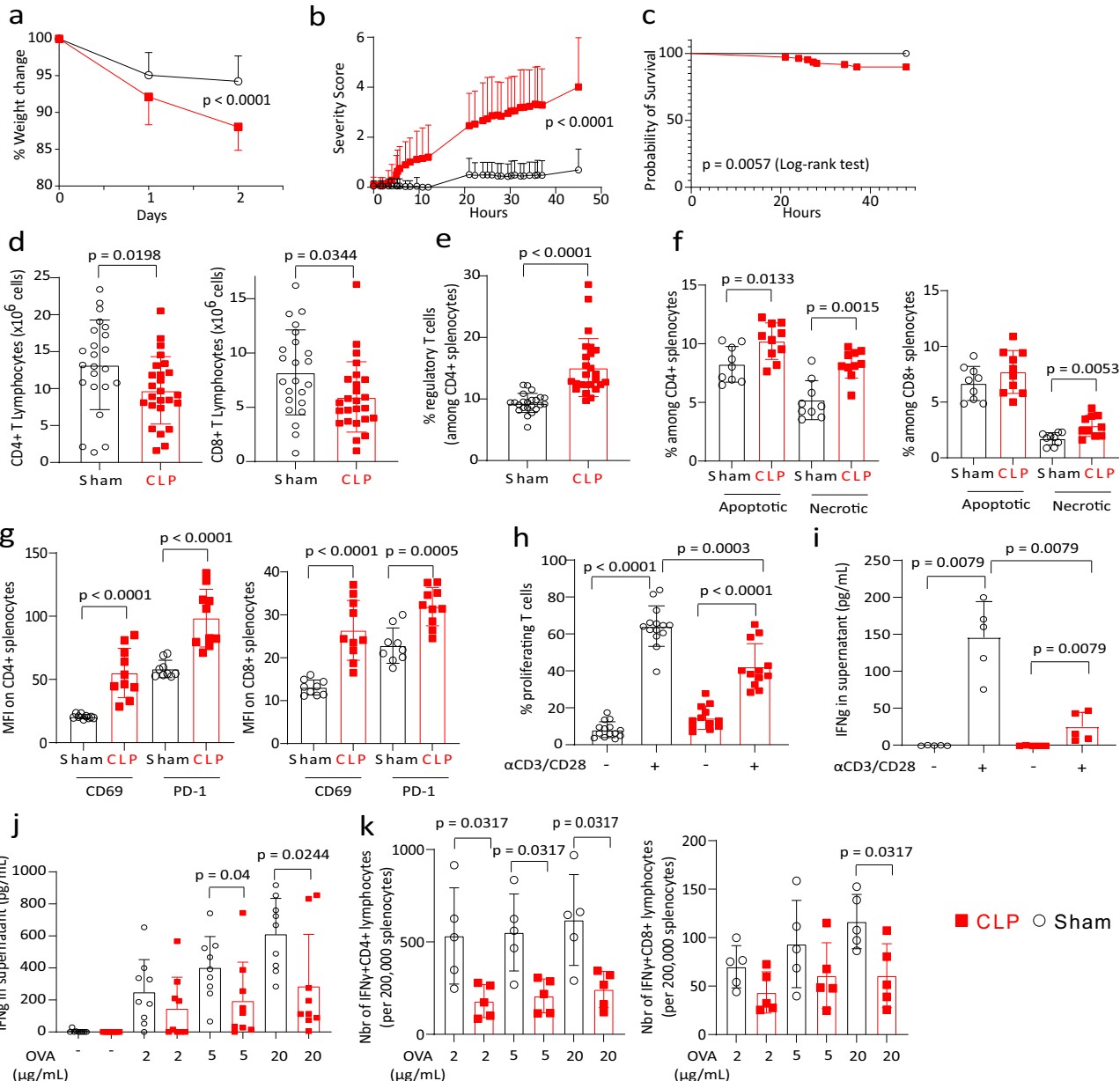

**Fig. 1 | Clinical and immunological features of the reanimated cecal ligation and puncture model. a** Percentage of weight change on day 1 and day 2 post-surgery compared to weight prior to surgery. **b** Clinical severity score measured every hour during 2 days post-surgery. **c** Kaplan–Meier survival curves. Significance in survival analysis was determined by log-rank test (**a–c**: $n = 72$ Sham vs 109 CLP). **d**, **e** Absolute counts of CD4+ T, CD8+ T, and percentages of CD4+CD25+CD127low regulatory T cell ($n = 23$ Sham and 25 CLP mice). **f** Percentages of apoptotic and necrotic cells among CD4+ T and CD8+ T lymphocytes ($n = 9$ Sham and $n = 10$ CLP). **g** Median Fluorescence Intensity (MFI) of PD-1 and CD69 on CD4+ T and CD8+ T lymphocytes ($n = 9$ Sham and $n = 10$ CLP). **h–i** Splenocytes from Sham ($n = 14$) and CLP ($n = 12$) mice were cultured with anti-CD3/CD28 antibodies-coated beads. **h** T-cell proliferation was measured among viable cells by flow cytometry. **i** IFNγ was measured in the supernatant of splenocyte cultures from Sham ($n = 5$) and CLP ($n = 5$) mice. **j**, **k** Splenocytes from ovalbumin-immunized mice were cultured with ovalbumin peptide at increasing concentrations (2, 5, 20 µg/mL). **j** IFNγ was measured in supernatants after 7 days ($n = 9$ Sham and $n = 9$ CLP). **k** Numbers of IFNγ producing CD4+ T and CD8+ T lymphocytes per 200,000 splenocytes were measured after 4 h ($n = 5$ Sham and $n = 5$ CLP). Red squares represent results in CLP mice and white circles in Sham animals. Geometric means with standard deviations (SD) and individual values are shown. Two-sided Mann–Whitney tests were performed, and only significant $p$ values < 0.05 are shown.

animals, Fig. 3i). Interestingly, *Ccr7*, *Cxcr5* and *Cxcr4* mRNA levels were decreased in CD138+ PCs purified from CLP mice suggesting that these cells may possess decreased homing predispositions (Fig. 3i).

Finally, co-culture experiments showed that proliferation (Fig. 3j) and IFNγ secretion (Fig. 3k) induced by TCR stimulation of CD3+ T splenocytes purified from Sham animals were significantly decreased in the presence of CD138+ PCs purified from CLP mice but not of PCs purified from Sham animals.

These results demonstrate that B220LowCD138+IgM+HLA-DR+ PCs are induced in the spleen after CLP and present with regulatory functions on T-cell proliferation ex vivo.

## In vivo depletion of CD138+ PCs improves T-cell proliferation after CLP
Next, we aimed to confirm in vivo that PCs induced after CLP participated in altered T-cell proliferation commonly observed after sepsis.

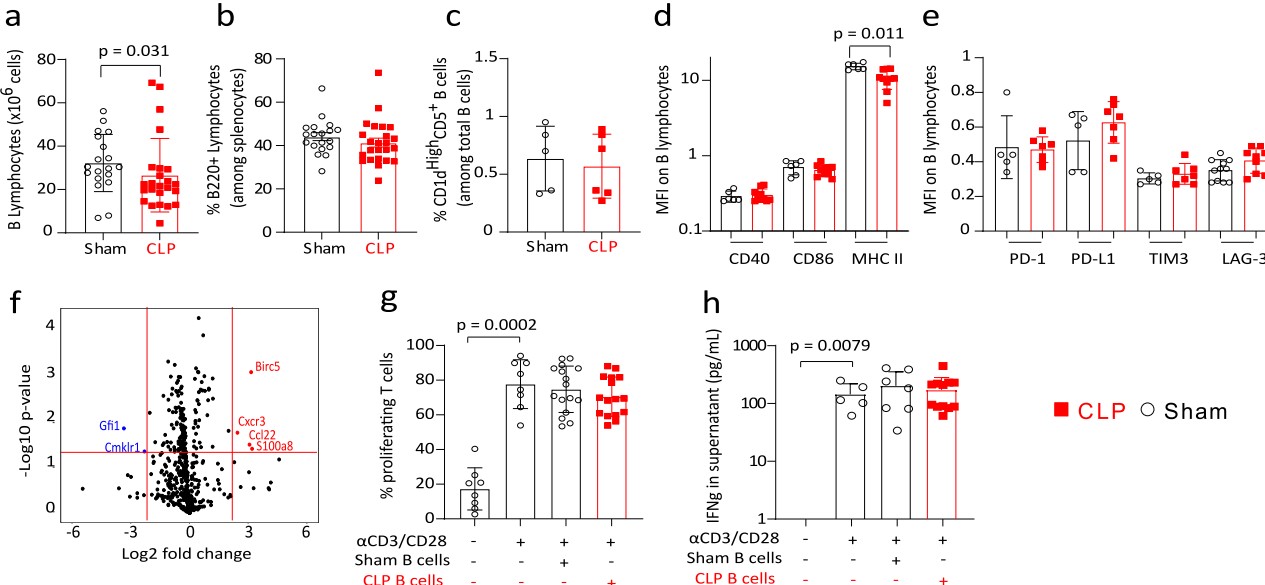

**Fig. 2 | Regulatory B lymphocytes are not induced in mice after CLP. a, b** B splenocytes counts and percentages in 19 Sham and 24 CLP mice. **c** Proportion of CD1d^highCD5+ B splenocytes in 5 Sham and 6 CLP mice. **d** CD40, CD86 and MHC class II expressions as median fluorescence intensity (MFI) on splenic B cells from 6 Sham and 10 CLP mice. **e** PD-1, PD-L1, TIM-3 and LAG-3 expressions as MFI on splenic B cells (*n* = 5 Sham and 7 CLP mice for all markers except for LAG-3 where *n* = 11 sham and 9 CLP). **f** Volcano plot showing fold changes of 540 mRNA expressions in splenic B cells from 6 Sham and 6 CLP mice. Significantly upregulated mRNA are shown in red and downregulated ones in blue. **g** Splenic T cells from Sham mice

(*n* = 8) were co-cultured with or without Sham (*n* = 16) or CLP (*n* = 16) purified splenic B cells at 1:1 ratio for 72 h, in the presence or absence of anti-CD3/CD28 antibodies-coated beads. T-cell proliferation was measured among viable cells by flow cytometry. Red squares represent results in CLP mice and white circles in Sham animals. **h** IFNγ was measured in culture supernatants of Sham T cells cultured either alone (*n* = 5) or with B cells from Sham (*n* = 7) or CLP (*n* = 12) mice. No IFNγ was detected in NS wells. Geometric means with standard deviations (SD) and individual values are shown. Two-sided Mann–Whitney tests were performed, and only significant p values < 0.05 are shown.

Thus we depleted CD138+ PCs in vivo through bortezomib administration after CLP (BZB, 0.5 mg/kg I.V.). BZB was administered 24 h after CLP in order to preferentially deplete PCs induced after sepsis *versus* resident PCs in spleen. Injection of dimethylsufoxyde (DMSO, i.e. BZB vehicle) was used as control condition in both CLP and Sham animals.

We confirmed that BZB administration induced a modest but significant depletion of PCs in the spleen after CLP (Fig. 4a, b) while the number of B220+ B splenocytes remained unmodified (Fig. 4c). BZB treatment was associated with restoration of the proliferative response of splenic T cells recovered from CLP mice as compared with that of splenic T cells originating from septic mice treated with DMSO (Fig. 4d). Finally, BZB administration led to a significant reduction in weight loss after 48 h in CLP animals compared with untreated septic animals (Fig. 4e).

Thus these results are in line with our hypothesis that CD138+ PCs induced after CLP participate in sepsis-induced impairment of T lymphocyte proliferative potential.

### CD138+ PCs induced after CLP regulate T-cell proliferation through cell-cell contact and IL-10 production

Next, we investigated putative pathways involved in the regulatory mechanism of CD138+ PCs induced after sepsis.

Transwell experiments were performed in order to evaluate whether the suppressive effect exerted by CLP-induced PCs on T-cell response relied upon direct cell-cell interactions. We observed that the absence of cell-cell contact between PCs and responder T cells was associated with a significantly increased T-cell proliferation as compared with co-culturing both cell populations in the same well (Fig. 5a). However, in Transwell condition, T-cell proliferation did not strictly return to control values thus suggesting that part of the immunoregulatory function of CLP-induced PCs could be mediated by the release of soluble factors.

As the central role of IL-10 in Breg and PCreg functions has been well described[18], we monitored CD44 expression on PCs after CLP,

which was documented as a marker of IL-10-producing B-cell subpopulations[19]. CD44 expression was significantly increased on CD138+ PCs after CLP (Fig. 5b), as observed by Matsumoto et al. in a model of experimental autoimmune encephalomyelitis[19]. We also evaluated IL-10 production in PCs induced after CLP in a model with *Il-10*-eGFP-reporter mice (VertX mice)[20]. We first confirmed that CD138+ PCs absolute numbers and percentages were increased in VertX mice after CLP as they were in wild-type animals (Fig. 5c, d). We next showed that IL-10-GFP expression was higher in PCs as compared with B cells both in Sham and CLP animals (Fig. 5e, f). In addition, IL-10 expression was significantly induced in CD138+ PCs after CLP compared with PCs from Sham animals (Fig. 5f). In accordance, IL-10 concentrations measured in the supernatants of CD138+ PCs purified from CLP mice were higher than in the supernatants of PCs purified from Sham animals irrespective of the presence of T cells in the culture (Fig. 5g).

In total, these results show that IL-10-producing CD44+ B220^LowCD138+IgM+ PCs are induced in spleen after CLP. Altogether, our data suggest that the immunoregulatory function of CLP-induced PCs is exerted both via cell/cell contacts and IL-10 release.

### Increased PD-L1 expression on CLP-induced CD138+ PCs participates in their immunoregulatory functions

Next, we evaluated putative cell surface receptors involved in the cell-cell contact regulatory mechanism of CD138+ PCs induced after sepsis.

Transcriptomic analysis of purified PCs showed that CLP induced a major transcriptional reprogramming of these cells with the differential regulation of 554 mRNAs compared with Sham animals (Fig. 6a). Seventeen pathways were significantly dysregulated in PCs after CLP (Fig. 6b). The majority of these pathways were downregulated (*n* = 16, 94%) after CLP such as the B cell receptor signaling or NF-kB signaling pathways.

Most importantly, only one pathway was upregulated in CLP-induced CD138+ PCs: the PD-1/PD-L1 immunoregulatory pathway

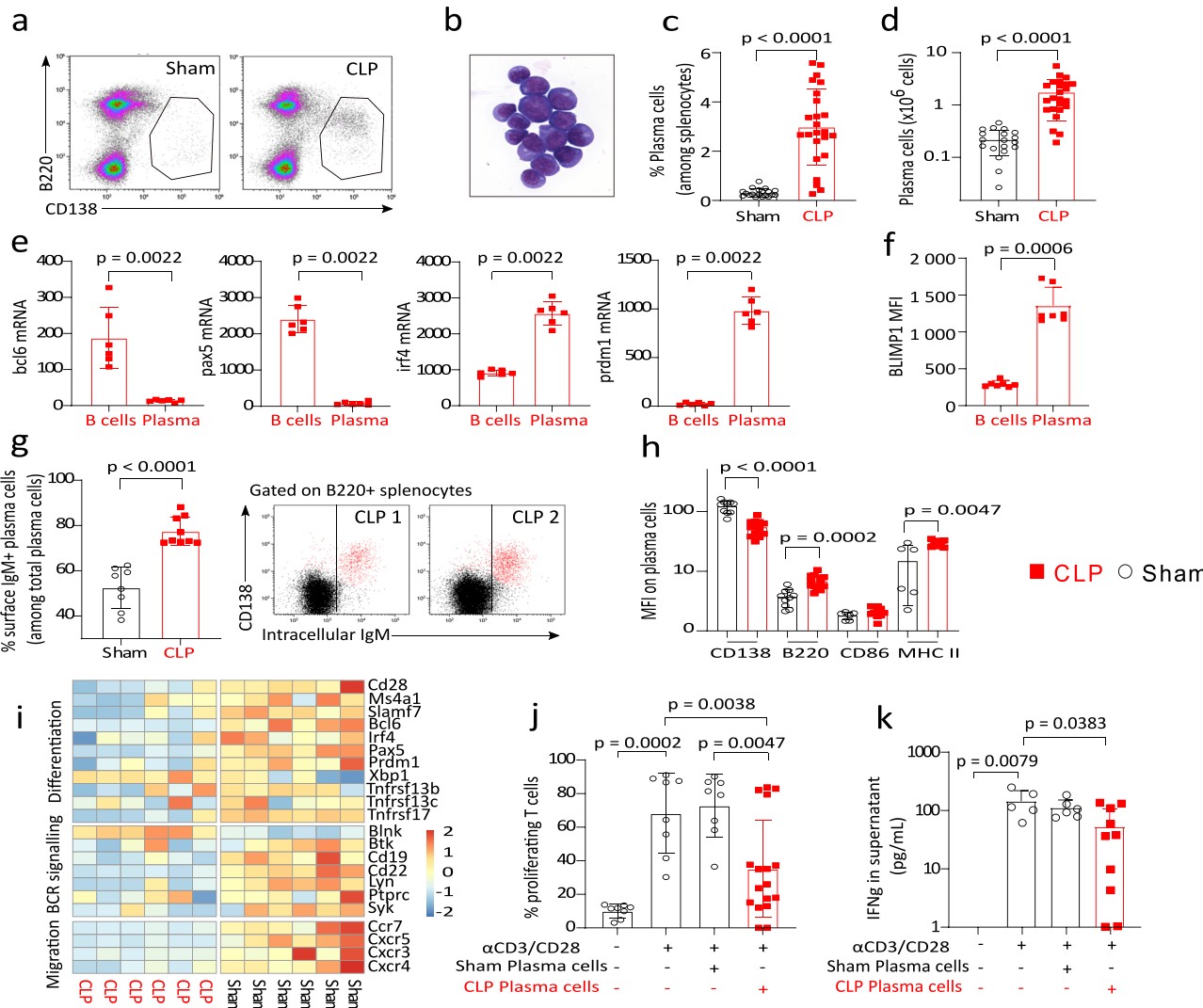

**Fig. 3 | Regulatory CD138+ IgM+ plasma cells are induced in spleen after CLP.**
**a** Representative examples of flow cytometry gating of plasma cells (PCs) in one Sham and one CLP animal. **b** Morphology of sorted CD138+ B220low splenic cells from a CLP mouse. **c, d** Proportions and absolute counts of PCs among splenocytes in 19 Sham and 24 CLP mice. **e** mRNA expressions of transcription factors *bcl6*, *pax5*, *irf4*, and *prdm1* (coding for BLIMP1) measured in B cells and PCs purified from 6 CLP mice. **f** BLIMP1 expression as median fluorescence intensity (MFI) on splenic B cells and PCs in 7 CLP mice. **g** Percentages of cell surface IgM-expressing PCs (among the entire PC population) in 8 Sham and 9 CLP mice. Representative examples of IgM intracellular expression profile in B220+ splenocytes from 2 CLP mice **h** Expressions on PCs in Sham and CLP mice of CD138 (*n* = 10 and 13), B220 (*n* = 10 and 13), CD86 (*n* = 6 and 10) and MHC II (*n* = 6 and 10) as median fluorescence intensities (MFI). **i** Heatmap of mRNA expressions of selected transcripts related to cell differentiation, BCR signaling, or migration measured in PCs purified from 6 Sham and 6 CLP mice. The red and blue colors indicate over- and under-expressed markers, respectively, and the color intensity specifies the expression level. **j** Splenic T cells from Sham mice (*n* = 8) were co-cultured with Sham (*n* = 8) or CLP (*n* = 17) splenic PCs at 1:1 ratio for 72 h, with or without anti-CD3/CD28 antibodies-coated beads. T-cell proliferation was measured among viable cells by flow cytometry. **k** IFNγ was measured in culture supernatants of Sham T cells cultured either alone (*n* = 5) or with PCs from Sham (*n* = 6) or CLP (*n* = 10) mice. Red squares represent results in CLP mice (either B cells or PCs), and white circles in Sham animals. Geometric means with standard deviations (SD) and individual values are shown. Two-sided Mann–Whitney tests were performed, and only significant *p* values < 0.05 are shown.

(Fig. 6b). This change primarily corresponded to the upregulation of the PD-1 Ligand, as illustrated by the increased expression of PD-L1 on PCs from CLP animals shown by flow cytometry analysis (Fig. 6c). After CLP > 90% of PCs were positive for PD-L1 expression (Fig. 6d). Conversely, LAG-3 expression, another immune checkpoint receptor expressed on PCs with regulatory functions[11], was not modified on CLP-induced PCs compared with Sham animals (Fig. 6e, f).

The functional consequences of upregulated PD-L1 expression on CLP-induced PCs were explored ex vivo by co-culturing splenic T lymphocytes with CLP-induced PCs, in the presence or absence of an anti-PD-L1 blocking antibody. As illustrated by Fig. 6g, the addition of the anti-PD-L1 blocking antibody induced a significant recovery of T-cell proliferation compared with the addition of the corresponding

isotypic IgG, thus suggesting that the PD-L1/PD-1 axis contributes to the immunosuppressive function of CLP-induced PCs.

Altogether, our results show that PD-L1+CD44+B220LowCD138+IgM+ PCs induced in the spleen after CLP present with inhibitory functions on T-cell proliferation involving both IL-10 production and upregulated expression of the checkpoint inhibitory ligand PD-L1.

## PD-L1+CD138+ PCs are induced in critically ill patients with bacterial sepsis and severe COVID-19 in association with increased mortality and regulate T-cell proliferation ex vivo

Next, we monitored the induction of PCs in 3 cohorts of ICU patients: one retrospective cohort of patients with bacterial sepsis (REALISM cohort)[21,22] and two prospective cohorts of patients diagnosed

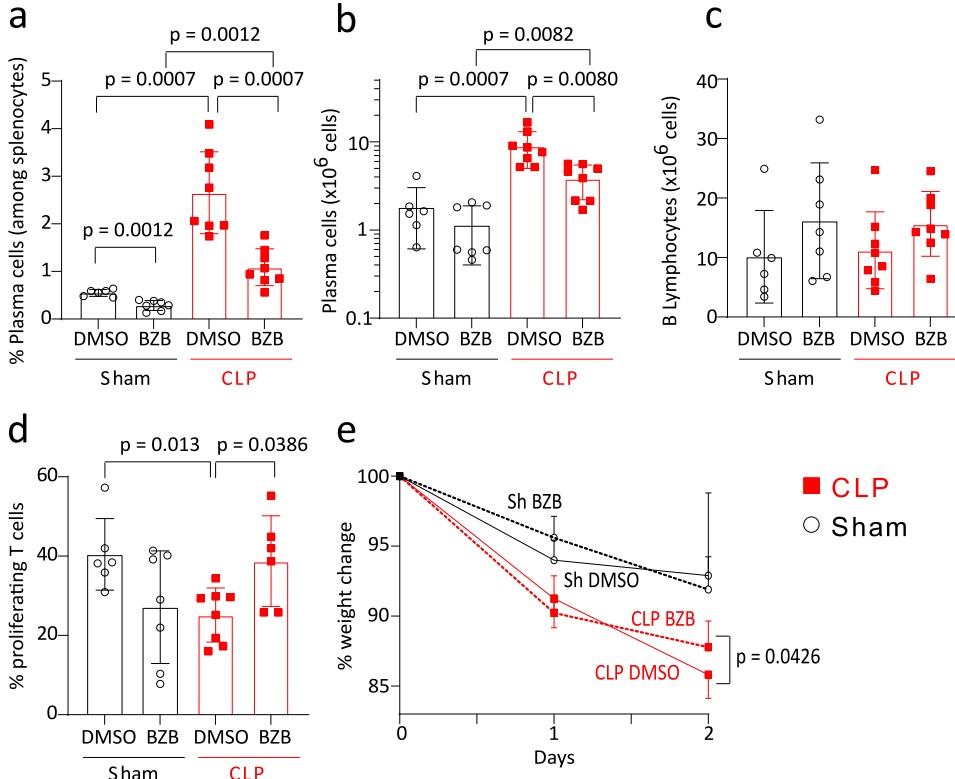

**Fig. 4 | In vivo depletion of CD138+ plasma cells improves T-cell proliferation after CLP.** Animals were either treated with Bortezomib ($n = 7$ Sham and 8 CLP mice) or DMSO (control condition, $n = 6$ Sham and 8 CLP mice) 24 h after surgery. **a** Proportions of plasma cells (PCs) among B cells and **b**, **c** absolute counts of PCs and B cells in spleen. **d** Splenic T cells from Sham and CLP mice were cultured with anti-CD3/CD28 antibodies-coated beads. T-cell proliferation was measured among viable cells by flow cytometry. **e** Percentage of weight change at day 1 and day 2 post-surgery compared to weight prior to surgery. Red squares represent results in CLP mice and white circles in Sham animals. Geometric means with standard deviations (SD) and individual values are shown. Two-sided Mann–Whitney tests were performed, and only significant $p$ values $< 0.05$ are shown.

respectively with bacterial sepsis (IMMUNOSEPSIS cohort)[23] and severe COVID-19 (RICO cohort)[24].

The induction of PCs in the blood of ICU patients with bacterial sepsis was first retrospectively evaluated by re-analyzing data from the REALISM study[21]. Demographic and clinical data for this cohort are presented in Supplementary Table 1. As CD138 and CD38 markers were not available in this cohort, flow cytometry stainings from the sub-cohort of septic patients and healthy donors were rea-analyzed to determine percentages and absolute counts of SSC$^{\text{Int}}$CD19$^{\text{low}}$ lymphocytes, previously demonstrated as a surrogate phenotype for circulating PCs[9]. The percentages of PCs were significantly increased in septic patients compared with healthy donors starting from D1 after ICU admission (Fig. 7a). Maximum value was observed one week (D5-D7) after ICU admission. PCs frequencies then progressively decreased over time until D60 without strictly returning to normal values. In this cohort, when comparing survivors and non-survivors at D14, we observed that the median PC frequency measured at D5 after ICU admission was significantly higher in non-survivors as compared with survivors (Fig. 7b).

These results were then validated prospectively in observational studies, including septic shock patients and critically ill patients with severe COVID-19 for which PCs were defined as CD138$^{+}$CD38$^{+}$CD19$^{\text{low}}$ lymphocytes. In addition, PD-L1 expression was monitored on circulating PCs. Demographic and clinical data for this cohort are presented in Supplementary Table 1. In patients with bacterial sepsis, we confirmed the significant increase in the percentage of CD138$^{+}$CD38$^{+}$ PCs compared with HV with a peak value at the end of the first week (Fig. 7c). A similar increase was observed in COVID-19 ICU patients. However, maximum value was detected earlier (i.e., upon ICU

admission, Fig. 7c). In both patients with bacterial sepsis and COVID-19, PCs presented with high PD-L1 expression (Fig. 7d, e), expressed intracellular IgM (Fig. 7f) and presented with elevated intracellular BLIMP1 expression (Fig. 7g). Thus, as observed in mice after CLP, PD-L1$^{+}$ PCs are induced in ICU patients after bacterial sepsis and COVID-19.

The immunoregulatory function of circulating PCs was assessed ex vivo for 8 critically ill infected patients (7 ICU patients with bacterial sepsis and 1 COVID-19). We observed that the addition of CD138$^{+}$ PCs purified from septic patients significantly reduced the proliferation of total T cells, of both CD4$^{+}$ and CD8$^{+}$ T cells and IFN$\gamma$ secretion induced by TCR proliferation of responder T cells purified from healthy donors (Fig. 7h–i, Supplementary Fig. 4A, B). By contrast, in 6 patients, the addition of B lymphocytes purified from septic patients had no effect on T-cell proliferative response and IFN$\gamma$ secretion in agreement with results observed in CLP mice (Fig. 7h–i).

Altogether, our data indicate that the induction of PCs with high PD-L1 expression and regulatory function on T-cell proliferation was observed in ICU patients with bacterial and viral sepsis (COVID-19) in association with increased mortality.

## CD138$^{+}$ PCs induction after sepsis correlates with that of regulatory T cells, but their immunoregulatory functions may be independent

Last, we asked whether immunoregulatory PCs induced after sepsis may require Treg to inhibit T-cell proliferation.

We observed that the absolute numbers of Treg and PCs were significantly correlated after sepsis both in patients (Fig. 8a) and in CLP mice (Fig. 8b). This suggests that common induction mechanisms may

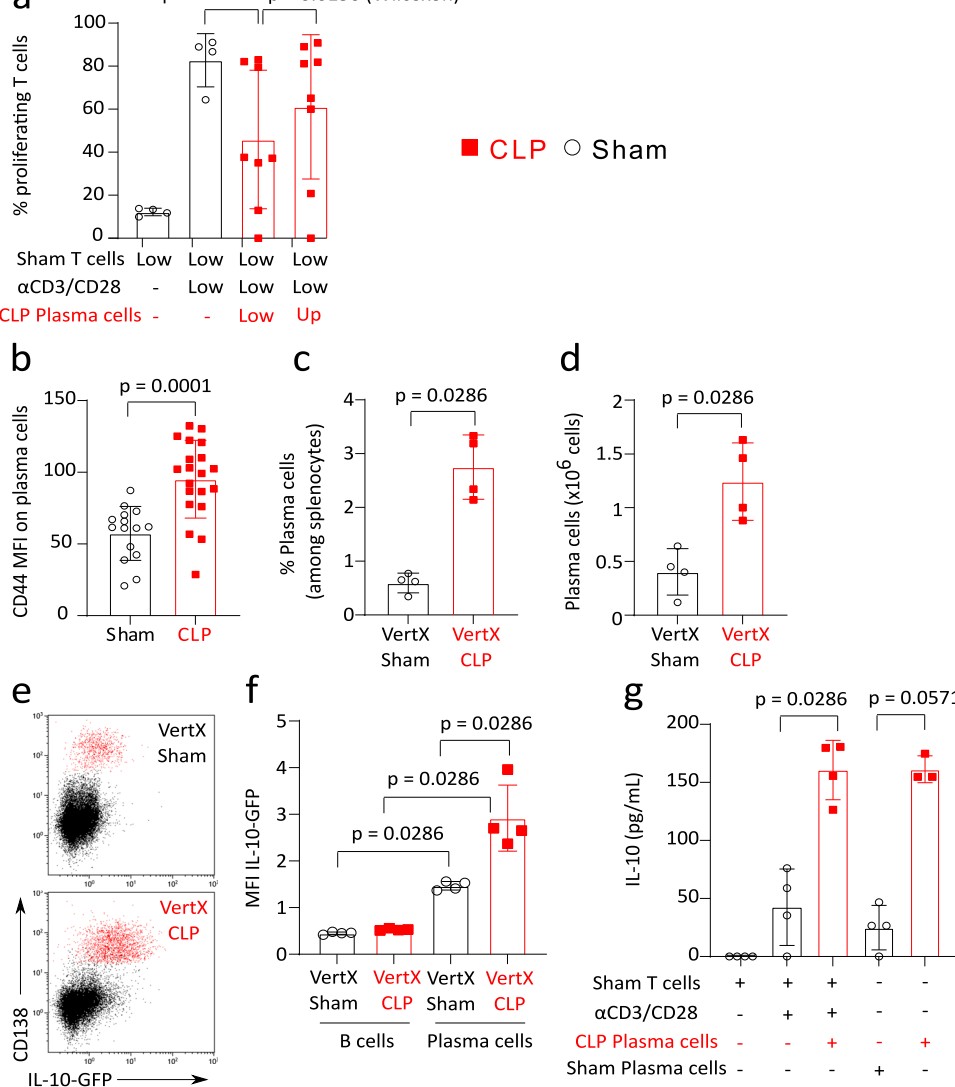

**Fig. 5 | CD138+ plasma cells induced after CLP regulate T-cell proliferation through cell-cell contact and IL-10 production. a** Splenic T cells from Sham mice ($n = 4$) were co-cultured with plasma cells (PCs) purified from CLP mice ($n = 8$) with or without HTS Transwell® inserts (T cells in the bottom of the well and PCs on the top of the membrane) at a 1:1 ratio for 72 h, and stimulated by anti-CD3/CD28 antibodies-coated beads. T-cell proliferation was measured among viable cells by flow cytometry. A non-parametric Wilcoxon paired test was used to compare proliferation in the presence of PCs with or without the inserts. **b** CD44 expression as median fluorescence intensity (MFI) on PCs measured in 15 Sham and 20 CLP mice. **c,d** Percentages among B cells and absolute counts of PCs in 4 CLP and 4 Sham Il-10-eGFP-reporter VertX mice. **e** Representative examples of IL-10 expression profiles in B cells and PCs (red dots) in Sham and CLP VertX mice. **f** IL-10 expression as median fluorescence intensity (MFI) in B cells and PCs from 4 Sham and 4 CLP VertX mice. **g** IL-10 production measured in culture supernatants of unstimulated Sham T cells ($n = 4$), Sham T cells stimulated by anti-CD3/CD28 antibodies-coated beads and co-cultured with or without CLP PCs ($n = 4$) or in supernatants of PCs purified from Sham ($n = 4$) or CLP mice ($n = 3$). Red squares represent results in CLP mice and white circles in Sham animals. Geometric means with standard deviations (SD) and individual values are shown. Two-sided Mann–Whitney tests were performed, and only significant $p$ values < 0.05 are shown.

lead to the differentiation/activation of both immunoregulatory cell types.

However, in BZB-treated mice, sepsis-induced PCs depletion and improvement in splenocytes proliferation was not associated with reduced Treg expansion (Fig. 8c).

Finally, ex vivo co-culture experiments showed that CD138+ PCs purified from septic patients were equally efficacious in decreasing T-cell proliferation in the presence or not of Treg in the co-culture (Fig. 8d–f).

In total, while these data suggest a common induction mechanism for Treg and PCs after sepsis, these immunoregulatory lymphocyte subsets may represent independent regulatory mechanisms to control T-cell proliferation. Nevertheless, as we cannot definitively rule out a

direct effect of BZB on Treg function, the confirmation of this last aspect would require a dedicated mouse model deficient for Treg.

## Discussion

A better understanding of the pathophysiology of sepsis-induced immune alterations is desirable to lead to novel therapeutic strategies to prevent and reduce the rates of secondary infections and associated mortality. Expansion of cellular subsets with immunoregulatory functions has been proposed as one of the mechanisms involved in this process. This has been particularly well described for CD4+CD25+Foxp3+ regulatory T cells and, more recently, for MDSCs[4]. However, the contribution of the more recently described regulatory

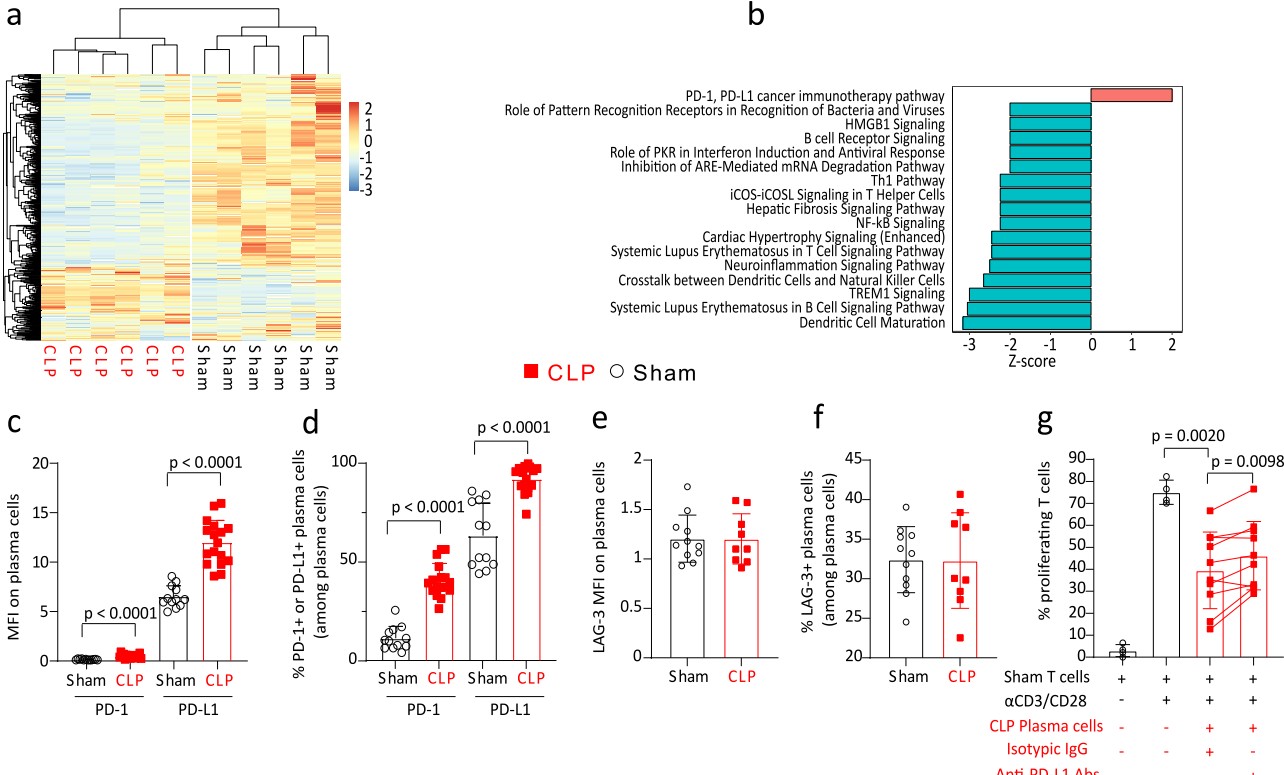

**Fig. 6 | CLP-induced PD-L1 expression contributes to plasma cells regulatory function. a** Heatmap of 554 mRNAs expressed in plasma cells (PCs) purified from spleens of 6 Sham and 6 CLP mice. The red and blue colors indicate over- and under-expressed markers, respectively, and the color intensity specifies the expression level. Hierarchical clustering was performed using Euclidean distance method. **b** Ingenuity Pathway Analysis of pathways significantly differentially regulated in PCs from CLP ($n = 6$) compared with Sham mice ($n = 6$). Upregulated and downregulated pathways are shown in red and blue, respectively. **c** PD-1 and PD-L1 expressions as median fluorescence intensity (MFI) on PCs measured in 12 Sham and 17 CLP mice. **d** Percentages of PD-1 and PD-L1 expressing PCs measured in 12 Sham and 17 CLP mice. **e**, **f** LAG-3 expression as MFI on PCs and percentages of PCs expressing LAG-3 among total PCs measured in 11 Sham and 9 CLP mice. **g** Splenic T cells from Sham mice ($n = 4$) were co-cultured with or without PCs purified from CLP animals ($n = 10$) at 1:1 ratio for 72 h with anti-PD-L1 blocking antibodies (10 μg/mL) or corresponding isotypic IgG (10 μg/mL), and stimulated or not by anti-CD3/CD28 antibodies-coated beads. Non-parametric Wilcoxon paired test was used to compare results obtained in these last 2 conditions. T-cell proliferation was measured among viable cells by flow cytometry. Red squares represent results in CLP mice and white circles in Sham animals. Geometric means with standard deviations (SD) and individual values are shown. Unless otherwise stated, two-sided Mann–Whitney tests were performed, and only significant $p$ values < 0.05 are shown.

cells of the humoral arm of the adaptive immune system (B cells and PCs) remains elusive in this context.

"Regulatory B cells" is a generic term that encompasses all B cells that act to suppress immune responses. While the phenomenology of B cell-mediated suppression is well established; the identity of B cells mediating these activities in vivo remains incompletely defined[25]. For example, in murine models of sepsis and in septic patients, the increased frequency of CD1d+CD5+ Breg[14,15], CD19+CD24highCD38high Bregs (i.e., B cells with a transitional phenotype)[26] and of CD27+CD24high B10 cells[27,28] has been described. Thus, there has been no consensus so far regarding the nature of the Breg subtype involved in the immunosuppressive phase of sepsis. In the current study, we observed neither induction of CD1d+CD5+ Breg, nor any strong modification of the B cell transcriptional profile or phenotype, nor acquisition by B cells of a regulatory function towards T cells after sepsis. In addition, in septic patients, we did not observe any regulatory effect of peripheral B cells on T-cell proliferation ex vivo. This argues against the induction of Breg in our model of sepsis. Besides differences in models and cohorts, the discrepancies between our present results and the published literature could be explained by 1/ the possibly concomitant existence and/or induction of several different regulatory B cell subpopulations in sepsis; 2/ the absence of any consensual phenotype characterizing regulatory B cells subsets and 3/ the limits of studying regulatory B cells solely based on phenotypic characterization in the absence of any functional exploration.

More recently, Nascimento et al. showed the expansion of splenic CD39highCD138high PCs in CLP mice, which participated in immunosuppression through increased extracellular adenosine production and inhibition of macrophages anti-microbial functions[29]. In their study, PCs expansion was established in a cohort of 21 septic patients. Here, we expand and complete these data by demonstrating that PD-L1+CD44+B220LowCD138+IgM+ regulatory PCs are induced in a murine model of sepsis and in critically ill septic patients. This was demonstrated not only by detailed analysis of their phenotypical features and gene expression profile but also by functional explorations comparing the capacity of purified B cells and PCs to suppress T-cell proliferation ex vivo. To our knowledge, this work is the first to strictly compare the regulatory potential of B cells and PCs in sepsis not only in mice but also in patients. In addition, our results extend the pathophysiologic role of PCreg in sepsis as besides their previously described inhibitory role on innate immune response[29], the current study demonstrates that these regulatory cells also participate in sepsis-induced impairment of T cell functions.

Interestingly, we describe here that the regulatory function of sepsis-induced PCs is dual in that it involves both secretion of a suppressive cytokine (IL-10) and expression of a member of the checkpoint inhibitory ligands family (PD-L1). This observation is in accordance with studies placing CD138high PCs as the major source of B cell-derived IL-10 in autoimmune, infectious and malignant diseases[19,30–33]. For example, Matsumoto et al. described IL-10-

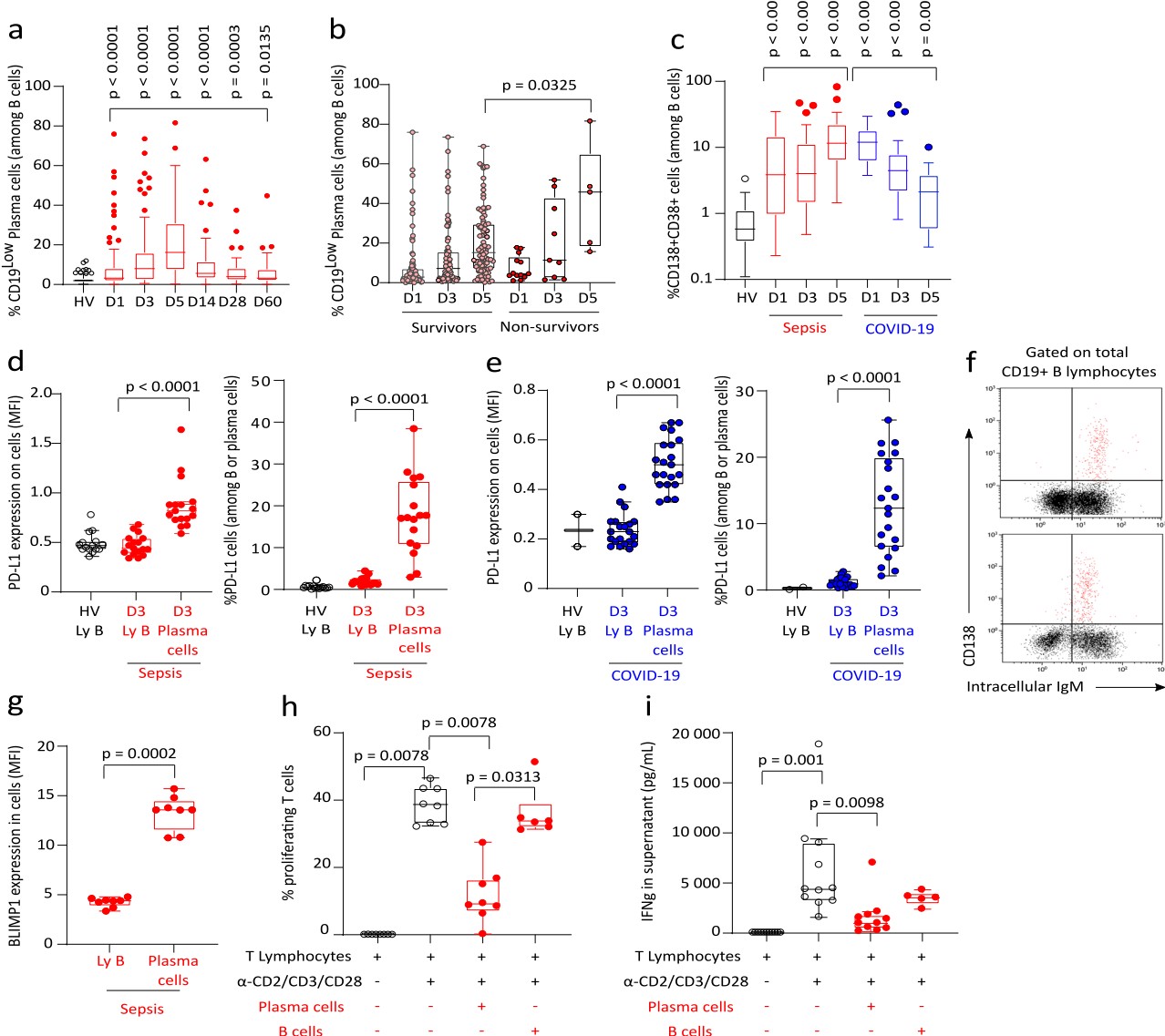

**Fig. 7 | PD-L1+ regulatory plasma cells in septic patients. a** Percentages of SSC[Int]CD19[low] plasma cells (PCs) in septic patients ($n = 107$) at D1 ($n = 92$), D3 ($n = 87$), D5 ($n = 92$), D14 ($n = 61$), D28 ($n = 46$), D60 ($n = 40$) and healthy volunteers (HV, $n = 175$). **b** Percentages of PCs between D14 survivors and non-survivors at D1 ($n = 79$ vs 13), D3 ($n = 78$ vs 9), D5 ($n = 87$ vs 5). **c** Percentages of CD38[+]CD138[+] PCs in septic ($n = 67$) and COVID-19 patients ($n = 33$) at D1 ($n = 41$ and 31), D3 ($n = 61$ and 20), D5 ($n = 40$ and 11) and HVs ($n = 33$). **d** Median fluorescence intensity (MFI) and percentage of PD-L1[+] B cells and PCs in 15 HVs and 17 septic patients at D3. **e** MFI and percentage of PD-L1[+] B cells and PCs from 2 HVs and 21 COVID-19 patients. **f** Representative histograms of CD138/IgM CD19[+]SSC[low] lymphocytes in 2 septic patients. Intracellular IgM staining was detected in >95% PCs (red dots). **g** BLIMP1 MFI in B cells and PCs from 8 septic patients. **h** Stimulated T cells proliferation ($n = 8$

HVs; white squares) with or without purified PCs ($n = 8$; squares) or B cells ($n = 6$, dots) from septic ($n = 7$) or COVID-19 patient ($n = 1$). **i** IFNγ in supernatants of HVs T cells cultured alone ($n = 11$) or with B cells ($n = 5$) or PCs ($n = 11$) from septic patients. Results are shown as box plots (box between first and third quartiles with horizontal line at median value, horizontal lines representing upper and lower adjacent values, values higher than the upper and lower adjacent values shown as dots). **b–e**, **g–i** individual values are also represented. Two-sided Mann–Whitney tests were performed to compare results obtained in patients versus HVs, and two-sided Wilcoxon paired tests to compare results of T-cell proliferation and IFNγ concentration between culture conditions. Only significant p values < 0.05 are shown.

producing CD138[+]CD44[+]HLA-DR[high] PCs that limit autoimmune inflammation in a murine model of multiple sclerosis[19]. Thus, besides autoimmune diseases, the current results expand the immunoregulatory role of CD138[+]CD44[+]HLA-DR[high] PCs as IL-10 producers and regulators of T-cell proliferation after sepsis as well. Our description of IL-10-producing PCs with a regulatory function is also coherent with the fact that IL-10 has been repeatedly described as a major mediator of sepsis-induced immunosuppression[34].

Besides IL-10, we demonstrated that the PD-1/PD-L1 axis is part of the PC regulatory armamentarium in sepsis inasmuch as PD-L1 expression is upregulated on sepsis-induced PCs and anti-PD-L1

blocking antibodies partially alleviated PC inhibitory function ex vivo. LAG-3, another inhibitory checkpoint ligand was identified by Lino et al. in a murine model of *Salmonella typhimurium* infection as a marker of a novel subset of regulatory PCs[11]. In addition, the authors showed that these cells also expressed high levels of other immune checkpoint ligands such as PD-L1, PD-L2 and CD200. The authors suggested that such co-expression of several receptors implicated in the negative regulation of immunity may be important as these molecules may act synergistically. However, the contribution of these receptors to the immunosuppressive function of PCs was not evaluated. Our current study shows that the inhibitory function of sepsis-

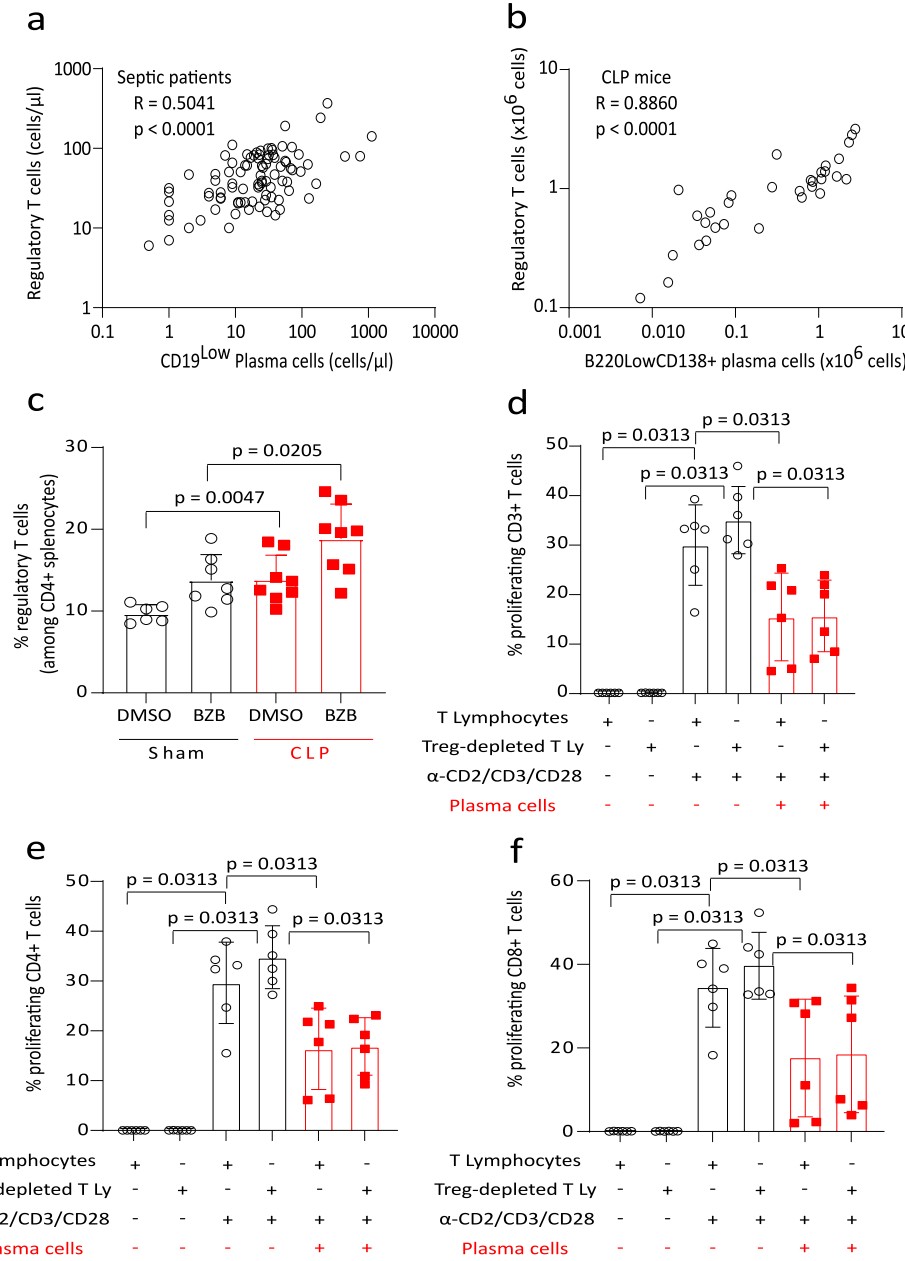

**Fig. 8 | CD138+ plasma cells and regulatory T cells after sepsis in mice and patients. a** Correlation between absolute counts of regulatory T cells and plasma cells (PCs) in the blood of patients at D5 after sepsis (*n* = 91). **b** Correlation between absolute counts of regulatory T cells and PCs in spleen of CLP mice (*n* = 32). Spearman correlation test was applied. R correlation coefficient and p values are shown. **c** Percentages of regulatory T cells among CD4+ splenocytes were measured in Sham (*n* = 7 and 6 respectively) and CLP (*n* = 7) mice treated with Bortezomib (BZB) or its control (DMSO). Red squares represent results in CLP mice and white circles in Sham animals. Geometric means with standard deviations (SD) and individual values are shown. Mann–Whitney tests were performed to compare the proportions of regulatory T cells in mice. **d–f** Circulating T cells from healthy volunteers (*n* = 6), depleted or not for regulatory T cells, were co-cultured with circulating PCs purified from septic patients (*n* = 6) at 1:1 ratio for 72 h and stimulated by anti-CD2/CD3/CD28 antibodies-coated beads. Proliferation of CD3+ **d**, CD4+ **e** and CD8+ **f** T cells was measured among viable cells by flow cytometry. Geometric means with standard deviations (SD) and individual values are shown. Two-sided Wilcoxon paired tests were used to compare results of T-cell proliferation between culture conditions. Only significant p values < 0.05 are shown.

induced PCs towards T cells is not mediated through increased LAG-3 expression but involves activation of the PD-1 pathway. Such suppressive function of PD-L1/PD-L2 has also been documented for IgA+ PCs involved in intestinal immune homeostasis[32,35]. Although Umakoshi et al. reported high levels of IL-10 and PD-L1 mRNA in CD1d+CD5+ Bregs from CLP mice[14], contribution of the PD-1/PD-L1 axis to the regulatory function of these cells was not addressed. However, treatments targeting the PD-1/PD-L1 axis are currently considered in septic patients[36], which might also help to lift part of the regulatory signals delivered by PCs in septic patients.

Finally, we studied the relationship between the two adaptive immunoregulatory lymphocyte subpopulations induced after sepsis, namely Treg and sepsis-induced immunoregulatory PCs. As observed in cancer patients[37,38], Treg and PCs were correlated and concomitantly induced both in patients and mice after sepsis, suggesting a common induction mechanism. However, we suggest that these immunoregulatory lymphocyte subsets may constitute independent regulatory mechanisms to control T-cell proliferation after sepsis. Indeed, we observed that Treg induction after CLP was maintained in BZB-treated mice. In addition, the immunoregulatory function of PCs on T-cell

proliferation ex vivo was preserved in Treg-depleted samples. To note, in these last experiments, the improvement of T-cell proliferation due to Treg depletion was close to statistical significance ($p = 0.0625$, non-parametric Wilcoxon paired test) as it was observed in all paired samples except for one. This is in agreement with previous data showing that different PC subpopulations produced IL-10 and IL-35 in a murine model of infection[32]. In contrary, in some experimental settings of autoimmune diseases (uveitis and experimental autoimmune encephalomyelitis), regulatory B cell subpopulations were shown to induce Treg[32,39] via increased IL-35 production[40]. Further work is now necessary to decipher the pathways induced after sepsis leading to the induction/activation of these immunoregulatory cell subsets. Such a study may have an important impact on sepsis as results could help to identify treatable traits in patients, impacting simultaneously several immunoregulatory mechanisms.

In line, our present study also brings two lines of evidence indicating that PCs are important contributors to sepsis-induced immune alterations and, therefore, constitute valid therapeutic targets to improve immune cell functions impaired by sepsis. First, we showed that elevated PCs frequencies are associated with increased mortality in septic patients. Second, we found in the murine CLP model that in vivo PC depletion through bortezomib administration is associated with decreased features of sepsis-induced immune alterations and improvement of clinical endpoints. While further studies are now mandatory to confirm the clinical rationale for targeting PCs in sepsis, definition of biomarkers of PCs regulatory function will facilitate monitoring these cells in patients. These will be necessary not only to identify patients who could benefit from specific immunotherapies in an individualized targeted therapeutic approach but also to follow response to treatments.

Beyond bacterial sepsis, we describe here the induction of PCs with regulatory functions in clinical samples from ICU patients with severe COVID-19. Expansion of PCs has been repeatedly described in the most severely ill patients. Some studies even reported PC frequencies exceeding 30% of the total B cell pool[10,41–43]. However, the putative regulatory function of these expanded PCs has never been evaluated so far. In addition, PCs expansion in COVID-19 was correlated with hallmarks of an extrafollicular B cells response activation[10]. Extrafollicular B cell responses help to curb viral infection by allowing rapid production of IgM-expressing PCs[44]. In accordance, PCs induced in our model presented with features of extrafollicular B cell response with high IgM expression and reduced *Cxcr5* and *Cxcr4* mRNA levels suggesting decreased follicular and bone-marrow homing predisposition. However, the link between extrafollicular B cell responses and the expansion of regulatory PCs in sepsis has yet to be determined.

The induction of PCs with regulatory functions in patients with both viral and bacterial sepsis suggests that common mechanisms may lead to their expansion in infected ICU patients. Ex vivo studies showed that several stimuli such as BCR signaling, CD40-CD40L interaction and TLR stimulation may be required to induce PCreg differentiation[45]. In keeping with this, systemic inflammation and pro-inflammatory cytokines release induced by TLR ligands have been described as one mechanism of regulatory PCs expansion[46]. Thus we postulate that systemic inflammation induced after bacterial and viral sepsis may participate in regulatory PCs differentiation. In agreement, we showed that PCs induction was transient after sepsis with a return to normal value one month after sepsis putatively in parallel with inflammation resolution in these patients[21]. It is noteworthy that the kinetics of PCs induction in patients with bacterial and viral sepsis were different with a maximum value upon ICU admission in COVID-19 patients and 3 days after admission in patients with bacterial sepsis. This could be related to differential kinetics of inflammation, to higher viral/antigen burden or just be the reflection of different elapsed time after infection onset in these two clinical contexts (i.e. on average 10 days after date of first symptoms in COVID-19 patients and within 48 h after diagnosis of

infection in patients with bacterial sepsis). Dedicated research projects are now mandatory to fully explore mechanisms leading to PCs expansion and differentiation after viral and bacterial sepsis.

The current study presents with limitations. We acknowledge some level of variability in the results generated in the murine model of sepsis. In addition to the intrinsic variability associated with this model of sepsis[47–49], this is related to the mode of results presentation which included in a single figure pooled data coming from multiple experiments performed over a lengthy period by different experimenters without excluding any outlier. We are convinced that, since statistically significant differences were observed between Sham and CLP groups despite this variability, this supports the robustness of these results.

In addition, we did not evaluate the long-term persistence of immune alterations following CLP in the current model. This was due to ethical constraints from the Animal Ethics Committee associated with the current design of the CLP model. Therefore, we could not formally demonstrate the development of delayed immunosuppression in this model. However, in septic patients, we showed that the sepsis-induced increase in circulating PCs persisted at least two months after the onset of sepsis and was associated with increased mortality. This thus supports a role for these cells in sepsis-induced immunosuppression in these cohorts.

Also, the link between extrafollicular B cell responses and the expansion of regulatory PCs in sepsis cannot be fully defined based on the current results. In addition, the independence between Treg and PCreg immunoregulatory functions must now be confirmed in dedicated Treg-deficient mouse models. Induction mechanisms leading to PC expansion and differentiation after viral and bacterial sepsis should be explored. Finally, biomarkers to specifically identify PCs with regulatory functions in the clinic should be developed. These will be necessary to confirm the clinical rationale for targeting PCs in sepsis.

In conclusion, we demonstrate the role of regulatory PCs in the pathophysiology of sepsis-induced immune alterations. We show that these cells are induced in a relevant murine model of sepsis and in clinical samples from patients, where they exert suppressive effects on T cells through IL-10 production and elevated PD-L1 expression. Specific biomarkers of regulatory PCs must be developed to enable the monitoring of these cells in clinical settings. Additionally, the therapeutic potential of targeting these regulatory cells in patients should now be further investigated and validated.

## Methods

### Experimental design: mice cohorts

Male C57BL/6 J mice (strain code = #632, IMSR_JAX #000664, Charles River, L'Arbresle, France) and VertX mice containing *Il10*-eGFP-reporter (IMSR_JAX #014530, The Jackson Laboratory, ME, USA), aged between 8 to 10 weeks and weighing from ~21 to 29 grams were used in this project. VertX mice presented with an internal ribosome entry site (IRES)-enhanced green fluorescent protein (eGFP) fusion protein placed downstream of exon 5 of the interleukin 10 (*Il10*) gene[20]. Mice were housed one week before the experiments in a conventional animal facility (Service Commun des Animaleries de Rockefeller, Lyon, France and a PBES-specific pathogen-free rodent facility (ENS Lyon)). For each experiment, mice (total $n = 295$) from the same cage were randomly assigned to two groups: animals with polymicrobial sepsis induced by CLP (total $n = 172$) and control mice (Sham, total $n = 123$). They were kept together after surgery and until euthanasia. Experiments were systematically repeated in a minimum of at least 2 independent batches, including Sham and CLP mice. Results from all available experiments were pooled within a single figure without excluding any outliers. For example, in Fig. 1a–e, all the results of mice follow-up and spleen cell immune monitoring available at the time of initial manuscript submission were presented. This represented a pool of data coming from nine different experiments performed between 2017 and 2020 by three different experimenters of both surgery and

bench work. All the experiments were approved by the Université Claude Bernard Lyon 1 animal ethical evaluation committee (#2016072616404065, and #45069-2023062011386592, CECCAPP, Lyon, France) in accordance with the European Convention for the Protection of Vertebrate Animals used for Experimental and other Scientific Purposes.

### Experimental design: patients' cohorts

Adult patients from intensive care units (ICU) of the academic hospital (Hospices Civils de Lyon, Lyon, France) enrolled in three observational clinical cohorts were studied either prospectively (RICO = REA-IMMUNO-COVID and IMMUNOSEPSIS 4 cohorts) or retrospectively (REALISM = REAnimation Low Immune Status Markers cohort)[22]. Samples in humans were collected under approved protocols registered under ClinicalTrials: NCT04392401 (RICO = REA-IMMUNO-COVID); NCT02638779 (REALISM cohort (REAnimation Low Immune Status Markers); NCT04067674 (IMMUNOSEPSIS 4).

RICO cohort included critically ill patients who presented with pulmonary infection with SARS-CoV-2 confirmed by RT-PCR testing. Exclusion criteria disqualified pregnant women and institutionalized patients. It was approved by the ethics committee ("Comité de Protection des Personnes Ile de France 1"-N°IRB/IORG #: IORG0009918) under agreement number 2020-A01079-30. IMMUNOSEPSIS 4 included septic shock patients admitted into ICU identified according to the Third International Consensus Definitions for Sepsis and Septic shock (Sepsis-3)[1]: vasopressors administration started within the first 48 hours around inclusion, plasma lactate level above 2 mmol/L and diagnosed or suspected infection. The exclusion criteria were any condition modifying the immune status: immunosuppressive treatment (including > 10 mg equivalent prednisone per day or cumulative dose >700 mg), hematological disease treated within the 5 years, solid tumor treated with chemotherapy or in remission, a number of circulating PNN < 500/mm3, innate immune deficiency, extra-corporeal circulation with one month before inclusion (cardiac surgery or ECMO). The onset of septic shock was defined by the beginning of vasopressor therapy. It was approved by the ethics committee ("Comité de Protection des Personnes Ouest II−Angers Sud-Est II"; RCB identification number: 2019-A000210-57, identification number SI/CPP: 19.01.23.71857), registered at the French Ministry of Research and Teaching (#DC-2008-509) and recorded at the Commission Nationale de l'Informatique et des Libertés. Since RICO and IMMUNOSEPSIS 4 studies were observational with low risk for the patients and no specific blood sampling procedure besides routine blood sampling required, there was no need for written informed consent, although oral information and non-opposition to inclusion in the study were mandatory and recorded in patients' clinical files. Finally, REALISM included septic patients identified according to the Third International Consensus Definitions for Sepsis and Septic shock (Sepsis-3)[1]: vasopressors administration started within the first 48 hours after ICU admission, plasma lactate level above 2 mmol/L in case of septic shock, suspected infection for which microbiological sampling had been performed, along with the administration of antimicrobials. Exclusion criteria were aplasia or pre-existent immunosuppression, pregnancy, and individuals with no social security insurance, restricted liberty or under legal protection. It was approved by the ethics committee ('Comité de Protection des Personnes Sud-Est II', Bron, France) and the French National Security Agency for drugs and health-related products (Approval code: 69HCL15_0379, 30th November 2015).

In all three cohorts, EDTA and heparin anticoagulated blood samples were collected in patients during routine procedures. First samples (i.e., D1) were obtained within the first 48 h after inclusions in the studies (i.e. on average 10 days after the date of first symptoms in COVID-19 patients and within 48 h after diagnosis of infection for patients with bacterial sepsis). Clinical and biological parameters were collected, including demographic characteristics, date and cause of admission to ICU, vital status at day 28 after inclusion, type of infection, comorbidities (Charlson index) and severity scores (Simplified Acute Physiology Score II, Sepsis-related Organ Failure Assessment; McCabe classification). Concomitantly, EDTA and heparin anticoagulated blood samples were collected from healthy volunteers (HV) from the blood bank EFS of Lyon (Etablissement Français du Sang). According to EFS standardized procedures for blood donation, informed consent was obtained from healthy donors and personal data were anonymized at the time of blood donation and before the transfer of blood to our research lab. Every experiment was performed on fresh whole blood collected on the same day. Samples were kept on ice during storage before analysis processing.

### Murine model of CLP

For each experiment, mice (total $n = 295$) were divided in two groups: animals with polymicrobial sepsis induced by CLP (total $n = 172$) as previously described[50] and control mice (Sham, total $n = 123$). Briefly, around 1 hour after a subcutaneous (SC) injection of 0.1 mg/kg buprenorphine (AXIENCE), mice were anaesthetized with isoflurane inhalation (induction 3%, maintenance 1.7–2%; FiO2 = 0.2). They were placed on a heating mat during the whole surgery procedure. The abdomen was shaved and disinfected. Local anesthesia was performed by SC injection of 100 μL of lidocaine chlorhydrate 1% (AGUETTANT). Paramedian laparotomy allowed cecum exteriorization, which was then ligatured at its external third using 5.0 silk thread (Teleflex, DEKNATEL) and punctured twice (21-gauge needle) without crossing to the other side. A droplet of feces was extruded from the puncture holes to ensure patency. Cecum was carefully replaced in the peritoneum to avoid any intestinal twist. Finally, incision was sutured in layers, and animals were resuscitated with an SC injection of 1 ml of Ringer Lactate (VIAFLO). Mice were placed under a heat lamp during recovery. Within the same conditions, Sham control mice underwent laparotomy with only exteriorization of cecum from abdomen cavity, without ligation nor puncture. Mice were kept in the same cage after surgery, a heating mat was placed under half of the cage, and food was softened with water. Six hours following surgery and then every 12 h for the next two days, all mice received injections of antibiotics (20 mg/kg/day intraperitoneal (IP) Metronidazole BRAUN; 85 mg/kg/day SC Cefoxitine PANPHARMA). Pain was controlled by SC injection of 0.1 mg/kg buprenorphine given 10 hours post-surgery, and twice daily for the next two days. 48 h after surgery, mice were anaesthetized with isoflurane before sacrifice by exsanguination from retro-orbital sinus (local anesthesia by tetracaine 1%) and cervical dislocation.

A severity score was evaluated every hour (except during the night) after surgery based on several clinical criteria: fur aspect, motor activity, posture, breathing, weight loss (Supplementary Table 2). Animals were euthanized if they reached a score ≥9 before the 2 days end-point.

### In vivo depletion of PCs

0.5 mg/kg of Bortezomib (SIGMA-ALDRICH) or a control solution of 1/300 diluted DMSO (SIGMA-ALDRICH), i.e. Bortezomib vehicle, was injected intravenously in both Sham and CLP mice 24 h after surgery. This dose of Bortezomib was determined after dose-response experiment evaluating effectiveness of Bortezomib injection (0.5, 0.75 and 1 mg/kg) at depleting PCs without adversely affecting other cellular populations in spleens of CLP and Sham mice.

### Cell purification

In mice, spleens were mechanically homogenized in phosphate buffer saline (PBS, EUROBIO) under sterile conditions to obtain a suspension of splenocytes. Red blood cell lysis was performed using a hypotonic solution of NaCl, and cells were then washed twice in PBS. PCs were purified from splenocytes by positive selection with magnetic beads (EasySep™ Mouse CD138 Positive Selection Kit; STEMCELL), while

B and T cells were purified by negative selection (EasySep™ Mouse B and T Cell Isolation Kits; STEMCELL). Purified cells were further used for proliferation assay or frozen at −80 °C in 300 μL RNA Lysis Buffer (ZYMO RESEARCH) for further RNA expression analysis.

In humans, HVs' T cells were isolated from whole blood by immunodensity isolation (RosetteSep™ Human T Cell Enrichment Cocktail; STEMCELL). When necessary, regulatory T cells were depleted from T cells by magnetic cell separation (EasySep™ Human Pan-CD25 Positive Selection and Depletion Kit; STEMCELL), with a yield of <1% residual CD25+ CD4+ T cells. In septic patients, B cells were purified from whole blood by immunodensity isolation (RosetteSep™ Human B Cell Enrichment Cocktail; STEMCELL) and PCs by magnetic cell separation (MACSprep™ Multiple Myeloma CD138 MicroBeads; MILTENYI).

### Analysis of cellular morphology by microscopy
Between 20,000 and 40,000 B cells and PCs were sorted by FACS (Aria II; BECTON DICKINSON) as B220+ CD138− and B220low CD138+ cells, respectively. Cytocentrifuge was then used to concentrate cells onto a microscope slide so that they could be stained by May-Grünwald Giemsa and examined by a histologist blind of cell sorting populations. Images were acquired with a Nikon Eclipse 80i microscope at 60X magnification using a Basler camera and ICS Capture software (Tribvn, France).

### Cell culture and proliferation assays
In mice, total splenocytes or splenic T cells were stained with Tag-it Violet™ Proliferation and Cell Tracking Dye (BIOLEGEND). T cell cultures ($10^6$ cells/mL) were set up in a culture medium consisting of RPMI (EUROBIO) supplemented with 10% AB human serum (Etablissement Français du Sang, Lyon, France), 200 mM L-glutamine (LONZA), 10 mg/mL penicillin-streptomycin (BIOLOGICAL INDRUSTRIES), 5 mg/mL fungizone (BRISTOL-MYERS SQUIBB) and 1% β-mercaptoethanol (SIGMA), in 96-well flat-bottom plates. Cells were stimulated by anti-CD3/CD28 antibodies-coated beads (T Cell Activation/Expansion Kit; MILTENYI) at a bead-to-cell ratio 2:1.5, or by phytohemagglutinin (4 ng/μL) (THERMOFISHER). In the co-culture assays, splenic B cells or PCs were stained with CellTrace™ Far Red (THERMOFISHER) and added at 1:1 ratio to T cells in the stimulated condition. After 72 h of culture, supernatant was frozen at −80 °C and cells harvested, then stained with propidium iodide (SIGMA-ALDRICH) or Zombie Aqua™ Fixable Viability Kit (BIOLEGEND) and analyzed on flow cytometer to determine proportion of viable proliferating T cells in each culture condition. To measure the proliferation of CD4 and CD8 T lymphocytes, cells were also stained with CD45-APC-Fire750, CD3-PE, CD4-APC and CD8-PC7 prior to analysis on a flow cytometer. To explore regulatory mechanisms of PCs, additional culture conditions were used. HTS Transwell® inserts (0.4 μm pore size polycarbonate membrane; CORNING) were set on 96-well plates, with T cells on the bottom of the well and PCs on the top of the membrane, at a cell-cell ratio 1:1 and a concentration of $10^6$ cells/mL. Anti-PD-L1 (Ultra-LEAFeaf Purified anti-mouse CD274, clone 10 F.9G2B7-H1; BIOLEGEND) blocking antibody or corresponding isotype control (Ultra-LEAF Purified Rat IgG2b kappa; BIOLEGEND) was also added in some cultures at 10 μg/mL.

In humans, T cell cultures ($10^6$ cells/mL) were set up in 96-well flat-bottom plates in the same culture medium as mouse splenic T cells but without β-mercaptoethanol. Between 50,000 and 100,000 purified T cells from HVs (depleted or not of Tregs) were cultured in the absence or in the presence of stimulating anti-CD2/CD3/CD28 antibodies-coated beads (T Cell Activation/Expansion Kit; MILTENYI) at a bead-to-cell ratio of 1.25:1, either alone or with B cells or PCs from septic patients at 1:1 ratio in the stimulated condition. After 72 h of culture, cells were harvested, stained with anti-CD3-APC-AF750, anti-CD4-APC and anti-CD8-KrO antibodies and T lymphocyte proliferative response was evaluated by flow cytometry in each condition

measuring 5-ethynyl-2'deoxyuridine (EdU) incorporation in T cells as described elsewhere (Click-iT© EdU Flow cytometry assay kit, INVITROGEN)[51].

### Mice immunization with ovalbumin and evaluation of antigen-specific T lymphocyte response ex vivo
Mice were immunized with subcutaneous injection in each hip of 0.1 mL of chicken ovalbumin peptide OVA$_{323-339}$ in complete Freund's adjuvant (OVA$_{323-339}$/CFA Emulsion at 1 mg/mL, HOOKE LABORATORIES). Mice were submitted to CLP or Sham surgery fourteen days later. Mice were sacrificed 48 h after surgery and spleens were harvested. Total splenic cells ($10^6$ cells/mL) were cultured in 96-well flat-bottom plates in complete culture medium in the absence or in the presence of OVA$_{323-339}$ peptide (HOOKE LABORATORIES) at 3 concentrations (20 μg/mL, 5 μg/mL and 2 μg/mL). The concentration of IFNγ in supernatants was measured by ELISA after 7 days of ex vivo re-stimulation. Intracellular IFNγ level in T cells was evaluated after 4 hours of ex vivo re-stimulation in the presence of brefeldin A (5.0 μg/ml, BIOLEGEND).

### Flow cytometry
In mice, total splenocytes absolute counts were determined using fluorescent microbeads (Flow-Count Fluorospheres, BECKMAN COULTER) and LDS 751 cell-permeant nucleic acid stain (INVITROGEN). For each immunophenotyping analysis, 100 μL of spleen cell suspension (i.e., 500,000 cells) were incubated with 10 μL of TruStain fcX™ antibody (clone 93; BIOLEGEND) for 10 min at 4 °C prior to immunostaining. Without washing step, cells were then incubated with 3 μL of each monoclonal antibodies (mAb) for 30 min in the dark at room temperature, then washed in PBS. List of reagents is provided in Supplementary Tables 3–8. Supernatant was discarded, and cell pellet resuspended in 300 μL of PBS and 1% formaldehyde (SIGMA-ALDRICH) before data acquisition on flow cytometer. For evaluation of necrosis and apoptosis, 500,000 splenic cells were stained with 100 μl of Zombie Violet™ Fixable Viability Kit (BIOLEGEND) diluted at 1:1000 in PBS during 15 min in the dark at room temperature. Cells were then washed and incubated with 20 μl of 1:20 diluted TruStain fcX™ antibody (clone 93; BIOLEGEND) for 10 minutes at 4 °C. After a washing step, cells were incubated with 20 μl of a mix containing 1:100 diluted mAb and 1:200 diluted Apotracker™ Green (BIOLEGEND) during 20 min in the dark at 4 °C. After a washing step, cells were resuspended in 300 μl of PBS before data acquisition on a flow cytometer. Necrotic cells were defined as positive for both Zombie and Apotracker and apoptotic cells as positive for Apotracker only. Representative histograms of the flow cytometry gating strategy are illustrated in Supplementary Fig. 5. Intracellular stainings for BLIMP1, IgM and IFNγ were performed using eBioscience™ staining buffer set (THERMOFISHER SCIENTIFIC) as followed: after last washing step, cell pellet was resuspended in 50 μL of fixation buffer and incubated for 30 min at 4 °C in the dark. Cells were then centrifuge, supernatant discarded before a washing step with 50 μL of permeabilization buffer. Cells were then incubated with monoclonal antibodies for 35 min at room temperature in the dark, then washed in perm buffer and finally resuspended in PBS before data acquisition on a flow cytometer.

In humans, for cell surface stainings, 100 μL of whole blood was washed twice with PBS. The cell pellet was resuspended in 100 μL of AB human serum and incubated with mAb for 15 min in the dark at room temperature. Red blood cells were then lysed with 500 μL Optilyse C (BECKMAN COULTER) for 10 min in the dark at room temperature. Cells were finally washed with PBS, and pellet resuspended with 300 μL PBS. Intracellular stainings for BLIMP1 and IgM were performed using PerFix-nc Kit (BECKMAN COULTER) as followed: after 15 min incubation of 50 μL of whole blood with membrane antibodies, cells were washed and supernatant discarded. Cells were next incubated with 25 μL of R1 reagent in the dark at room temperature and then washed.

Intracellular antibodies were diluted in 300 μL of R2 reagent and added to cells. After 30 min incubation at room temperature in the dark, cells were washed with 3 mL of R3 reagent (1×) and finally resuspended in R3 reagent before data acquisition on flow cytometer. In a retrospective study of REALISM data, PCs were identified as CD19$^{low}$ SSC$^{high}$ circulating cells, while in RICO and IMMUNOSEPSIS prospective studies, they were defined as CD38$^+$ CD138$^+$ cells. We previously showed that these two phenotypes were well correlated and relevant for PCs analysis in human blood[9].

All cytometry experiments were performed on Navios cytometer (BECKMAN COULTER) or LSR II (BD BIOSCIENCES), the same day as sample collection. Calibration beads (Flow-Set and Flow-Check Fluorospheres; BECKMAN COULTER) were run daily to check for routine alignment and day-to-day and long-term performance validation. Fluorescence-minus-one or negative cell controls were used to establish positive windows when needed. FACS analysis used Kaluza software 2.1 (BECKMAN COULTER) and FlowJo 10.8.1 software (BD BIOSCIENCES). Depending on the marker or cell population analyzed, percentage and/or median fluorescence intensity (MFI) were monitored. Representative histograms of flow cytometry stainings expressed as MFI are illustrated in Supplementary Figs. 6–9.

### RNA extraction and gene expression analysis

Minimum 50,000 purified splenic B and PCs from 6 Sham and 6 CLP mice (i.e., 24 samples in total) were thawed for total RNA extraction, using RNeasy Plus Mini Kit (QIAGEN) following manufacturer's protocol. RNA concentration and integrity number (RIN) were measured with 2100 Bioanalyzer System (AGILENT). For each sample, 100 ng RNA presenting with RIN between 8.6 and 10 were processed for gene expression analysis. NANOSTRING nCounter assay, a hybridization-based multiplex assay, was performed for gene expression analysis at the hospital genomic platform (Plateforme BIOGENET Sud−Hospices Civils de Lyon, France). Total RNA were diluted with Nuclease-Free water (SIGMA-ALDRICH) at 20 ng/μl in the 12-strip provided by NANOSTRING, and were analyzed using the nCounter® PanCancer Immune Profiling Panel according to manufacturer's instructions. Eight negative probes and six serial concentrations of positive control probes were included in the multiplexed reaction, respectively, allowing to determine the background signal and to perform a normalization to correct potential sources of variation associated with the technical platform, as described elsewhere[52]. Then, to prevent the effect of potential RNA input differences, we normalized the data according to the expression of two genes (Hdac3 and Cyld), identified with geNorm method[53] as stable between the four groups of samples studied. Finally, among 768 genes available in the nCounter® PanCancer Immune Profiling Panel, 110 turned out to have no expression in any of the 24 samples analyzed, and we selected genes with RNA expression in >75% of B cells or PCs samples for further analysis, which represented 540 genes for B cells and 554 genes for PCs.

### IL-10 measurement

IL-10 was measured in undiluted culture supernatants using sandwich ELISA MAX™ Deluxe Set Mouse IL-10 (BIOLEGEND) according to manufacturer's recommendations. Samples were analyzed as duplicates when possible. Absorbance was read at 450 nm with Elx 808JU microplaque reader (BIOTECK). Gen5 software 3.04 (BIOTECK) was used to generate a standard curve through regression analysis on the log of standard values and to calculate samples' concentrations.

### IFNγ measurement

In mice, IFNγ was measured in undiluted culture supernatants using Mouse IFN-gamma DuoSet ELISA (BIO-TECHNE) according to manufacturer's recommendations. Samples were analyzed as duplicates. In humans, IFNγ was measured in culture supernatants using Human IFN-gamma DuoSet ELISA (BIO-TECHNE) according to

manufacturer's recommendations. Samples were analyzed undiluted for non-stimulated culture conditions or after 1/50 dilution for stimulated culture conditions, as duplicates when possible. Absorbance was read at 450 nm with ELx808 microplaque reader (BIOTECK). Gen5 software 3.04 (BIOTECK) was used to generate a standard curve through a four-parameter logistic curve-fit and to calculate samples' concentrations.

### Statistical analyses

Experimental data were analyzed by R (Version 3.6.2; Boston, MA, USA) and GraphPad Prism (Version 9.0.2; La Jolla, CA, USA, Supplementary Table 9). Statistical details of the experiments are provided in the respective figure legends. Bar plots present individual values as well as mean and standard deviation. Comparison of continuous variables was performed with non-parametric Mann–Whitney or Wilcoxon tests for matched samples. Kaplan–Meier survival analysis and log-rank test were used to compare mortality in Sham and CLP mice. RNA expression in purified B and PCs from mice was explored drawing volcano plots and heatmaps (pheatmap package in R), and using Ingenuity Pathway Analysis (QIAGEN). $P$ values < 0.05 were considered statistically significant and are shown on plots.

### Reporting summary

Further information on research design is available in the Nature Portfolio Reporting Summary linked to this article.

## Data availability

Source data are provided in this paper.

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

## Acknowledgements

The authors would like to thank (i) the clinical research assistants from ICUs of Edouard Herriot Hospital (Valérie Cerro, Laurie Bignet, Marion Provent among others) for their great help in the screening and sampling of patients as well as in the preparation of case report forms, (ii) the technical staff from the Edouard Herriot Hospital Immunology Lab for help in sample management (iii) bioMérieux joint research unit for

dedicated involvement in flow cytometry and transcriptomic data analysis (iv) Center d'Investigation Clinique de Lyon (CIC 1407 Inserm) for help in RICO clinical study management. The authors would also like to acknowledge the important contribution and motivation of the medical and paramedical staff of the Edouard Herriot Hospital. Finally, the authors would like to thank the patients and their families for supporting this study. REALISM study received funding from the Agence Nationale de la Recherche through a grant awarded to BIOASTER (Grant number #ANR-10-AIRT-03) and from bioMérieux, Sanofi and GSK. IMMUNO-SEPSIS study was funded by Hospices Civils de Lyon. RICO study was supported by funds from the Hospices Civils de Lyon, Fondation HCL and Claude Bernard Lyon 1 University/Région Auvergne Rhône-Alpes. This work was supported by funds from grant ANR-20-CE92-0024-01 to BP and FV, ERC-2013-CoG_616986 to BP, IUF to F.V. C.V. is a recipient of a Poste d'Accueil Inserm. We acknowledge the contribution of SFR Biosciences (UMS3444/CNRS, US8/Inserm, ENS de Lyon, UCBL) PBES rodent facility and ANIRA flow cytometry facilities, as well as of the Etablissement Français du Sang Auvergne—Rhône-Alpes.

## Author contributions

Conceptualization: M.G., T.D., O.T., G.M., F.V. Methodology: M.G., C.V., T.D., O.T., G.M., F.V., B.F.P. Investigation: A.V., R.C., L.J., J.L., K.K., M.O., H.P., M.M., A.F., E.P., C.G., M.B., A.C.L., T.R., L.A., M.C., C.V., M.L., L.C., RICO, REALISM. Visualization: M.G., T.D., L.J., J.L., M.M., O.T., G.M., F.V., C.V., M.L. Funding acquisition: G.M., F.V. Project administration: M.B., F.V. Supervision: T.D., G.M., F.V. Writing–original draft: M.G., F.V. Writing–review & editing: M.G., C.V., T.D., A.V., M.L., L.C., A.C.L., R.C., H.P., L.J., J.L., K.K., M.O., M.M., A.F., E.P., C.G., M.B., T.R., L.A., M.C., O.T., G.M., F.V., B.F.P.

## Competing interests

M.M. and E.P. are employees of bioMérieux SA, an in vitro diagnostic company. R.C., M.M., E.P., C.G., A.C.L., T.R., and G.M. work in a joint research unit, co funded by the Hospices Civils de Lyon and bioMérieux. This private company has no role in study design, results analysis or publication. The remaining authors declare no competing interests

## Additional information

## REALISM study group

Sophie Arnal[8], Caroline Augris-Mathieu[8], Frédérique Bayle[8], Liana Caruso[8], Charles-Eric Ber[8], Asma Ben-Amor[8], Anne-Sophie Bellocq[8], Farida Benatir[8], Anne Bertin-Maghit[8], Marc Bertin-Maghit[8], André Boibieux[8], Yves Bouffard[8], Jean-Christophe Cejka[8], Valérie Cerro[8], Jullien Crozon-Clauzel[8], Julien Davidson[8], Sophie Debord-Peguet[8], Benjamin Delwarde[8], Robert Deleat-Besson[8], Claire Delsuc[8], Bertrand Devigne[8], Laure Fayolle-Pivot[8], Alexandre Faure[8], Bernard Floccard[8], Julie Gatel[8], Charline Genin[8], Thibaut Girardot[8], Arnaud Gregoire[8], Baptiste Hengy[8], Laetitia Huriaux[8], Catherine Jadaud[8], Alain Lepape[10], Véronique Leray[8], Anne-Claire Lukaszewicz[3,8], Guillaume Marcotte[8], Olivier Martin[8], Marie Matray[8], Delphine Maucort-Boulch[11], Pascal Meuret[8], Céline Monard[8], Florent Moriceau[8], Guillaume Monneret[1,3], Nathalie Panel[3], Najia Rahali[3], Thomas Rimmele[3,8], Cyrille Truc[8], Thomas Uberti[8], Hélène Vallin[3], Fabienne Venet ⓘ[1,2]✉, Sylvie Tissot[3], Abbès Zadam[3], Sophie Blein[4], Karen Brengel-Pesce[4], Elisabeth Cerrato[4], Valérie Cheynet[4], Emmanuelle Gallet-Gorius[4], Audrey Guichard[4], Camille Jourdan[4], Natacha Koenig[4], François Mallet[4], Boris Meunier[4], Virginie Moucadel[4], Marine Mommert[4], Guy Oriol[4], Alexandre Pachot[4], Estelle Peronnet[4], Claire Schrevel[4], Olivier Tabone[4], Julien Textoris[4], Javier Yugueros Marcos[4], Jérémie Becker[12], Frédéric Bequet[12], Yacine Bounab[12], Florian Brajon[12], Bertrand Canard[12], Muriel Collus[12], Nathalie Garcon[12], Irène Gorse[12], Cyril Guyard[12], Fabien Lavocat[12], Philippe Leissner[12], Karen Louis[12], Maxime Mistretta[12], Jeanne Moriniere[12], Yoann Mouscaz[12], Laura Noailles[12], Magali Perret[12], Frédéric Reynier[12], Cindy Riffaud[12], Mary-Luz Rol[12], Nicolas Sapay[12], Trang Tran[12], Christophe Vedrine[12], Christophe Carre[13], Pierre Cortez[13], Aymeric De Monfort[13], Karine Florin[13], Laurent Fraisse[13], Isabelle Fugier[13], Sandrine Payrard[13], Annick Peleraux[13], Laurence Quemeneur[13], Andrew Griffiths[14], Stephanie Toetsch[14], Teri Ashton[15], Peter J. Gough[15], Scott B. Berger[15], David Gardiner[15], Iain Gillespie[15], Aidan Macnamara[15], Aparna Raychaudhuri[15], Rob Smylie[15], Lionel Tan[15] & Craig Tipple[15]

[10]Hospices Civils de Lyon, Lyon-Sud University Hospital, Intensive Care Department, Pierre-Bénite, France. [11]Hospices Civils de Lyon, Service de Biostatistique et Bioinformatique, Pôle Santé Publique, Lyon, France. [12]BIOASTER Technology Research Institute, Lyon &, Paris, France. [13]Sanofi Pasteur, Marcy l'Etoile, France. [14]Laboratoire de Biochimie (LBC), ESPCI Paris, PSL Université, Paris, France. [15]GSK, GSK Medicines Research Centre, Stevenage, United Kingdom.

## RICO study group

**Remi Pescarmona[1], Lorna Garnier[1], Christine Lombard[1], Magali Perret[1], Marine Villard[1], Valérie Cheynet[4], Filippo Conti[4], Marie Groussaud[7], Marielle Buisson[7], Laetitia Itah[7], Inesse Boussaha[7], Françoise Poitevin-Later[1], Christophe Malcus[1], Eleonore Micoud[1], Florent Wallet[10], Marie-Charlotte Delignette[16], Frederic Dailler[17], Marie Simon[9], Auguste Dargent[9], Pierre-Jean Bertrand[9], Neven Stevic[9], Marion Provent[9], Laurie Bignet[8], Valérie Cerro[8], Jean-Christophe Richard[18], Laurent Bitker[18], Mehdi Mezidi[18] & Loredana Baboi[18]**

[16]Hospices Civils de Lyon, Croix-Rousse University Hospital, Anesthesia and Critical Care Medicine Department, Lyon, France. [17]Hospices Civils de Lyon, Pierre Wertheimer Hospital, Neurological Anesthesiology and Intensive Care Department, Lyon, France. [18]Hospices Civils de Lyon, Croix-Rousse University Hospital, Medical intensive Care Department, Lyon, France.

