## [Transparent Peer Review file · Nature Communications]

PD-L1 mediated T cell inhibition by regulatory plasma cells induced after sepsis

Corresponding Author: Professor Fabienne Venet

Version 0:

Reviewer comments:

Reviewer #1

(Remarks to the Author)

Major concerns:

In Figure 2, the authors show a slight change in the absolute number of B cells in the spleen of CLP mice, how about their percentage amongst splenocytes? Can the authors clarify the scale of the axes in panels C and D. Why is the MFI only below 10?

In Figure 3, the authors show that plasma cells express mainly IgM in the surface. By definition, plasma cells do not express Igs in the surface. Hence the authors should perform the same staining intracellularly and compare it to the current extracellular staining.

In Figure 7, the authors show how the peak of plasma cells in COVID-19 patients is at D1 compared to the sepsis condition where it is around D5. Can the authors clarify how the diagnosis of Sepsis and COVID-19 was performed and if the samples were taken only after receiving the positive diagnosis. This is an important aspect as the diagnosis can be delayed for one of the two conditions making the peak appear staggered. In figure 6, could the authors include the percentage of PD-L1+ cells in addition to the MFI? The authors should increase the number of samples in panel F to ensure enough statistical power.

The authors should confirm the plasma cell phenotype of the cells increased in both the CLP model and human samples with co-staining of Blimp1 and CD138 to confirm these are bona fide PCs. Indeed, there can be cells that express CD138 but are not Blimp1 positive.

The data shown in this paper seems to point to Bregs as the leading cause of sepsis-induced immunosuppression. However it is not clear if it Bregs are functioning in synergy with Tregs or independently. A model where Tregs are not present or depleted would be ideal to clarify the respective roles of Bregs and Tregs.

MFIs throughout the paper are inconsistent, with most MFIs being incredibly low, apart from in Figure 6C. The authors should address and correct if necessary, and give a plausible explanation for this inconsistency in the MFIs throughout the manuscript. The authors should provide representative histograms showing the difference in MFI together with each bar graph where MFI is presented.

Minor concerns:

In Figure 1, the authors show how the Treg cell percentage increases among CD4 T cells in CLP mice. If Tregs are increased, which compartment within CD4 T cells has their percentage decreased? Similarly, the authors show that absolute numbers of CD4 and CD8 T cells change between sham and CLP mice, but how about their percentages among live splenocytes? Similarly, how about the percentage and absolute number of NK cells, and myeloid cells such as DCs, pDCs, macrophages and monocytes?

In Figure 4, do the mice in this figure also show differences in disease scores (as in Figure 1B) after treatment with Bortezomib?

In Figure 6, the authors show the MFI of PD-1 and PD-L1 is increased in the PC of CLP mice. Does the percentage and absolute numbers of both PD-1+ and PD-L1+ PC show a similar trend?

Reviewer #2

(Remarks to the Author)

Gossez et al., investigate the induction of regulatory plasma cells after sepsis and COVID-19, and their mechanism of T cell suppression (measured as proliferation of bulk, non-specified, T cell subsets). Using cecal ligation and puncture (CLP) model in mice they show that the CD1dhiCD5+ regulatory B10 cells are not affected post CLP, however the B220lowCD138+IgM+ regulatory Plasma cells (reg-PCs) accumulate after CLP mediated sepsis. Using ex-vivo T cell/reg-PC co-culture experiments they show that these reg-PCs suppress T cell proliferation in response to CD3/CD28 stimulation. Finally, they show that these reg-PCs use potentially both cytokine (IL-10 production) and cell-cell contact (PD-L1) mediated mechanisms to suppress T cell proliferation. They also found that the human bacterial sepsis and COVID-19 sepsis patients have increased representation of CD138+ reg-PCs that exhibit increased PD-L1 expression, and consequently suppress T cell proliferation ex-vivo.

A key gap in knowledge is addressed here – to mechanistically define cellular mechanism(s) that control sepsis-induced state of immunosuppression (or immunoparalyses). The conceptual strength is noted – to use mouse sepsis model to guide and inform limited exploration in human septic and/or COVID-19 (viral sepsis) patients to define and characterize the role of reg-PCs.

Overall, the study hinted at potentially interesting mechanism that could contribute to the sepsis-induced T cell dysfunction.

However, few major limitations are noted that significantly diminished enthusiasm of this manuscript in its current stage.

Major Issues:

- The beauty of experimental mouse models is that variability in measures is relatively small enabling clear delineation and distinction between experimental groups/conditions... etc... It is puzzling where noted variability in a B6 mice comes from? Throughout the figures variability in control (sham) B6 mice is unexpectedly high – example Fig1D-F (ie. there is a mouse with 5x10⁶ cells per spleen and mouse that has 20x more...). This should be explained and ramifications of that discussed with the data and conclusions drawn throughout.
- Sepsis-induced immunosuppression is long-lasting phenomenon that changes over time in septic survivors. Extending analyses post first few days after sepsis induction (weeks and months) should be considered.
- T cell dysfunction is assessed here only through ability of T cells to proliferate in response to polyclonal stimulus. No attempts were made to define subsets of cells that are responding (and/or suppressed) – are they CD4 or CD8 T cells, are they naïve or effector or memory.... Many models are readily available to analyze the ability of Ag-experienced T cells to respond to their cognate antigens and provide effector functions - cytokine/chemokine production, killing capacity, protection to challenge/re-challenge afforded. These Measures are necessary to define T cell suppression.

Some of additional comments for authors to consider:

- The authors state that the regulatory PCs increase for up to 7 days in humans post bacterial sepsis and wanes after that. However, the kinetics of the reg-PCs and reg-B10 cells in mice is lacking and important in Fig.1. Providing this data can also help authors to justify the reason for selecting D2 for reg-PC characterization.
- While the authors show in Fig.2 (A, B) that the reg-B10 cell %/# are not affected post CLP, they didn't test their T cell suppression potential. In Fig.2 F, the authors co-culture T cells with total B cells, however, the same experiment with the CLP reg-B10 cells will inform the readers about the contribution of both reg-B10 cells and reg-PCs.
- In Fig 4, the authors conclude that the in vivo depletion of CD138+ reg-PCs improves T cell proliferation post CLP. While Bortezomib (BZB) is known to promote apoptosis in short- and long-lived PCs, this is still not a targeted depletion of CD138+ reg-PCs. Does Bortezomib influence re-B10 cell or other unknown Breg numbers and function, that can contribute to the observed result? Performing the sham and CLP surgery in *sd-1*^{-/-} (CD138-ko) mice or administering anti-CD138 antibody in vivo one day post CLP could be a better experiment to strengthen the authors claims.
- The improvement in weight loss observed in BZB treated CLP mice in comparison to DMSO treated CLP mouse is modest at best after 2 days post CLP (Fig 4.E). Collecting the weight loss and survival data for up to a week or longer with BZB could give a better significance to the authors claims.

Version 1:

Reviewer comments:

Reviewer #1

(Remarks to the Author)

The authors have successfully addressed most of the points raised in the first submission of this manuscript. However still a few new points, and some remain to be addressed:

Discussion points

New figure 8:

Bortezomib can decrease CD4 T cells in bone marrow and spleen (doi: 10.1371/journal.pone.0135081). Hence, despite the claim that the Tregs are increased in plasma cell-depleted mice with bortezomib, it cannot be ruled out that this drug, a proteasome inhibitor, could have an effect on Treg effector function and not number, hence the observed results on T cell proliferation. The authors should acknowledge that the situation in vivo could be different despite the results in vitro shown in panels D, E and F. A Treg-deficient KO mouse model would definitively and undoubtedly answer this question. In addition, according to the author's new data, the presence of Tregs seems do not affect the proliferation of T cells. Indeed, comparison of 2nd of 4th column of panels D, E and F is as it seems not significant. How can the authors explain Tregs are not suppressive when they should be? Please include this in the discussion

Figure 6G and Figure S3:

The results presented in these two figures have shared conditions (first and second column). However, they do not show the same levels (axes are also different). The results of these experiment should be carried out together and displayed together in a main figure.

Figure 7:

Authors should use the same type of graph in panels H and I as in panel G. Like this the individual data points will be observable.

As nicely done for the mice in Figure 3, have the authors measure IFN γ in the mice of figure 4?

Figure S2-B (right panel) is missing groups labels.

A better readout than weight loss to assess acute sepsis response could be temperature and/or IL-1b, IL-6, TNF or CRP measurement in mouse. Have the authors considered these readouts?

Line 228: authors mention Figure S3, but it should be figure S4.

Reviewer #2

(Remarks to the Author)

The authors provided detailed responses to major comments raised. However, one could argue that all 3 concerns were not adequately addressed.

1) Mouse to mouse variability is significant – much more than is published and available in the extensive literature on the subject. Sham mice are immunologically naïve, inbred, sex and age matched and anesthesia and/or surgery are not supposed to lead to those differences (20X differences in cellularity). Also, experiments performed here should not be considered as complex in design – they are rather simple and straightforward.

2) Despite limitations that they might have (15% weight loss) this manuscript is not defining sepsis induced immunosuppression using mouse CLP model.

3) Many models are readily available to analyze the ability of Ag-experienced T cells to respond to their cognate antigens and provide effector functions - cytokine/chemokine production, killing capacity, protection to challenge/re-challenge afforded. These measures are necessary to define T cell suppression. Those models are not utilized here - instead bulk, polyclonal stimulation is used which is limited in scope and reach.

Version 2:

Reviewer comments:

Reviewer #1

(Remarks to the Author)

I think that despite not fully addressing the concerns, the authors have provided acceptable explanation and have discussed the limitation of this manuscript.

I don't have any concerns.

Reviewer #2

(Remarks to the Author)

Version 3:

Reviewer comments:

Reviewer #2

(Remarks to the Author)

Significant amount of work is provided in response to the questions remained.

The manuscript is improved and limitations that are present are now properly discussed.

Point-by-point response to the reviewers' comments

First, we would like to thank the Reviewers for their thorough review of our manuscript and their thoughtful comments and suggestions. Below, you will find a point-to-point response to all points raised. We believe that our manuscript has improved considerably due to the review process. We provide also a revised version of the Manuscript in which changes are highlighted in red.

Yours sincerely,

Fabienne Venet, on behalf of the other authors.

Reviewer #1

Major concerns

In Figure 2, the authors show a slight change in the absolute number of B cells in the spleen of CLP mice, how about their percentage amongst splenocytes? Can the authors clarify the scale of the axes in panels C and D. Why is the MFI only below 10?

Response: We would like to thank this Referee for this question.

As requested, the percentage of B cells in the spleen of CLP mice is now shown in revised Figure 2 (Fig. 2B). While the absolute number of B cells was reduced in the spleen after CLP, we did not observe any difference in the percentages of total B220+ splenocytes between CLP and sham animals. This is now mentioned in the text (Page 7).

In former panels C and D in Figure 2, results were expressed as arbitrary units of medians of fluorescence intensities (MFI). This referee is correct that, in former panels C-D (current panels D-E) MFIs were below 10. These apparently low values are due to our experimental set-up.

In particular, it is well known that, in flow cytometry, MFI measurements are highly dependent on several experimental aspects

- Flow cytometer set up (voltage, compensation, ...)
- Flow cytometry staining protocol
- Reagents used (antibody clone, antibody dilution, tandem dye and fluorochrome brightness, ...)
- And most importantly flow cytometry software used by data analysis

To illustrate this last aspect, we provide below the example of one single flow cytometry file extracted from current study re-analyzed using 2 different softwares (Kaluza and FlowJo). Calculated MFI ranged from 0.66 (using Kaluza, i.e. results presented in Fig 2) to around 5 (when using FlowJo).

Software	Kaluza	Flow Jo
Median Florescence Intensity	0.66	5.32

Thus, we are convinced that the low MFI values presented in several figures of this article are related to the experimental setting of our flow cytometry studies. However, this does not preclude the quality of the presented data as appropriate controls were systematically included in each experiment. In addition, as requested, we provide representative histograms of each FCM staining expressed as MFI to illustrate the observed differences (or lack of) in Fig S6, Fig S7, Fig S8.

In Figure 3, the authors show that plasma cells express mainly IgM in the surface. By definition, plasma cells do not express Igs in the surface. Hence the authors should perform the same staining intracellularly and compare it to the current extracellular staining.

Response: This Reviewer is correct that we presented cell surface staining for IgM expression on plasma cells. This is because studies from our team [Blanc *Nat Com* 2016] and other [Pinto *Blood* 2013] previously demonstrated that a subpopulation of IgM-expressing plasma cells maintain a functional BCR (as opposed to IgG expressing plasma cells). These studies showed that these IgM+ plasma cells can sense antigen and acquire competence for cytokine production upon antigenic challenge.

With that said and as requested, we have performed intracellular IgM stainings in plasma cells from CLP mice (n = 13) and septic patients (n = 8).

We showed that plasma cells induced after sepsis expressed intracellular IgM both in patients and in mice. This is now mentioned in the text (Pages 8 and 13) and representative histograms are shown (Figure 3G and 7F).

In Figure 7, the authors show how the peak of plasma cells in COVID-19 patients is at D1 compared to the sepsis condition where it is around D5. Can the authors clarify how the diagnosis of Sepsis and COVID-19 was performed and if the samples were taken only after receiving the positive diagnosis. This is an important aspect as the diagnosis can be delayed for one of the two conditions making the peak appear staggered.

Response: We fully agree with this comment and this is an important question regarding comparison between critically ill patients with COVID-19 and bacterial sepsis.

In the REA-IMMUNO-COVID cohort (RICO clinical study, NCT04392401) including critically ill patients with SARS-CoV-2 infection, diagnosis of COVID-19 was based on a positive RT-PCR test for SARS-CoV-2 infection in at least one respiratory sample. Inclusions were made when patients were admitted to the ICU. On average, we estimated that the first sample (i.e. D1) was obtained between 10 and 15 days after the date of the first symptoms (start of the infection).

IMMUNOSEPSIS 4 study (NCT04067674) included septic shock patients identified according to the diagnostic criteria of SEPSIS 3 definition [Singer *JAMA* 2016]. The onset of septic shock was defined by the beginning of vasopressor therapy. The first sample (i.e. D1) was obtained within the first 48h after diagnosis of shock.

Thus, this difference in the delay between onset of infection and first sample may explain the difference in the kinetic of Preg that we observed between bacterial and viral sepsis (i.e. D1 sample of COVID-19 patients would correspond to D5 sample in septic patients). This is now mentioned in the Material and Methods (Page 23) and Discussion part of the Manuscript (Page 20).

In figure 6, could the authors include the percentage of PD-L1+ cells in addition to the MFI?

Response: As requested the percentages of PD-L1 expressing plasma cells are now included on Figure 6 (Panel D). We observed a significantly increased percentage of PD-L1+ plasma cells in CLP mice compared with sham animals (as observed with MFIs). After CLP, >90 % of plasma cells were positive for PD-L1 expression. This is also mentioned in the text (Page 11).

The authors should increase the number of samples in panel F to ensure enough statistical power.

Response: As requested, we have increased the number of experiments in former panel F of Figure 7 (now panel H). A total of 8 critically ill septic patients have now been included. We confirmed that addition of CD138⁺ PCs purified from septic patients to responder T cells purified from healthy donors significantly reduced their proliferation induced by TCR proliferation. By contrast, in 6 patients out of 8, addition of B lymphocytes purified from septic patients had no effect on T cell proliferative response in agreement with results observed in CLP mice. The Result part has been updated accordingly (Page 13).

The authors should confirm the plasma cell phenotype of the cells increased in both the CLP model and human samples with co-staining of Blimp1 and CD138 to confirm these are bona fide PCs. Indeed, there can be cells that express CD138 but are not Blimp1 positive.

Response: We would like to thank this Referee for this question. We have confirmed the plasma cell phenotype by several methods.

First, in the CLP model, we monitored the mRNA levels of transcription factors regulated during plasma cell differentiation in B cells and CD138⁺ plasma cells purified from spleens of CLP animals. As expected [Blanc *Nat Com* 2016] mRNA levels of *bcl6* and *pax5* were higher in B cells than in plasma

cells whereas mRNA levels of *irf4* and *prdm1* (coding for Blimp1) were lower in B cells than in plasma cells. Results are now presented in Figure 3E and discussed in the text (Page 8).

Second, we performed intracellular stainings for BLIMP1 in plasma cells from both septic patients and mice. When comparing the MFI of BLIMP1 between plasma cells and B cells from the same sample, we showed that plasma cells expressed higher BLIMP1 expression than B cells both in patients and mice. These results are now presented in Figure 3F (mice data) and Figure 7G (human data).

The data shown in this paper seems to point to Bregs as the leading cause of sepsis-induced immunosuppression. However it is not clear if it Bregs are functioning in synergy with Tregs or independently. A model where Tregs are not present or depleted would be ideal to clarify the respective roles of Bregs and Tregs.

Response: We appreciate this Reviewers' comment regarding the putative synergy between Preg and Treg in inducing sepsis-induced immunosuppression. Several observations and additional experiments can be offered.

First, we observed a simultaneous expansion of Treg and Preg after sepsis as the absolute counts of both cell types were significantly correlated in septic patients and CLP mice. This suggests that common induction mechanisms may lead to the differentiation / activation of these adaptive immunoregulatory lymphocytes. Deciphering these mechanisms would be of major interest in the better understanding of the pathophysiology of sepsis-induced immunosuppression and in the development of innovative therapeutic targets to prevent these induced immune alterations.

Second, we explored Treg response in plasma cell-depleted mice with bortezomib. In this experimental setting, while the percentages and absolute counts of Preg decreased in bortezomib-treated mice, the percentages of Treg measured in the same animals increased in parallel. However in this experiment, despite this increase of Treg, the proliferation of splenocytes was improved in BZB-treated mice after CLP.

Third, we performed Treg-depletion experiments in *ex vivo* co-culture tests with plasma cells from patients. We observed that Treg depletion from T cells of healthy donors did not impact Preg regulatory effect on T cells. In other words, Preg were equally efficacious in decreasing T cell proliferation in the presence or not of Treg in the co-culture.

In total, while these data suggest a common induction mechanism for Treg and Preg after sepsis, they show that these 2 immunoregulatory subsets may provide independent regulatory mechanisms to control T cell proliferation aimed at maintain immune response homeostasis.

These data are now presented in the Results (Page 14), Discussion (Page 18) as well as in new Figure 8.

MFIs throughout the paper are inconsistent, with most MFIs being incredibly low, apart from in Figure 6C. The authors should address and correct if necessary, and give a plausible explanation for this

inconsistency in the MFIs throughout the manuscript. The authors should provide representative histograms showing the difference in MFI together with each bar graph where MFI is presented.

Response: Again we appreciate this Reviewer's comment regarding the consistency of MFI throughout the manuscript.

Thanks to this remark, we have identified a problem with MFI values presented in Fig 6C; which were plotted 2 log higher than actual measured values. Fig 6C has thus been corrected and we would like to apologize for this mistake. This explains the discrepancy and inconsistency of MFI values noticed by this Referee.

Regarding the low values of MFI that are reported here, as discussed earlier in the text, we would like to emphasize that the MFI measurements by flow cytometry are highly related to experimental and instrumental conditions and in particular staining protocol set-up (clone of antibody used, selection of fluorochrome conjugates, antibody dilution, RBC lysis or not) and instrument set-up (voltage, compensation, software used for analysis, etc.). Therefore intrinsic values of MFI cannot be interpreted per se but only within the context of a single design of experiments with identical technical and instrument set-ups and by comparison with each other. For example, MFI cannot be compared between different staining protocols or targeted proteins.

We would like to reassure this Referee that, for each cell surface marker monitored by MFI in this study, protocol and instrument set-ups were strictly controlled so as to be able to rigorously compare MFIs between relevant conditions.

Indeed, 1/ the correct and reproducible functioning of the flow cytometer itself was evaluated daily through running of control beads (Flow Check and Flow set beads) to validate stability of fluidics and optic systems. 2/ Flow cytometry staining protocols for each panel were minutely set-up. For example antibody dilutions and numbers of cells per well were controlled and optimized and were similar between experiments. 3/ Appropriate staining controls were included (i.e. unstained cells, isotype controls or FMOs when appropriate). Finally, 4/ in each set of mice, control and septic mice were systematically present and a limited number of experiments were performed to limit the inter-experiment variability of the data.

To illustrate the robustness of our flow cytometry data, representative stainings for each FCM result expressed as MFI are now shown in supplementary figures (Fig S6, S7 and S8).

In closing, to illustrate our involvement in such standardization process of MFI measurement by flow cytometry, we can mention that our group has been pioneering in the standardization of the monitoring of decreased expression of HLA-DR on monocytes by flow cytometry as a biomarker of sepsis-induced immunosuppression. This marker involves the measurement and standardization of HLA-DR MFI on monocytes [Hamada S Cytometry B Clin Cytom. 2022, Bourgoin P Shock. 2021, Monneret G Cytometry B Clin Cytom. 2016, Demaret J Cytometry B Clin Cytom. 2013, Venet F Crit Care. 2011, Venet F Methods Mol Biol. 2011, Döcke WD Clin Chem. 2005, Finck ME Ann Biol Clin (Paris). 2003]. Using a standardized protocol, we have defined normal values for this marker which are now used for

patients' stratification in multicentric international interventional studies in sepsis [IGNORANT, IMMUNOSEP].

Minor concerns:

In Figure 1, the authors show how the Treg cell percentage increases among CD4 T cells in CLP mice. If Tregs are increased, which compartment within CD4 T cells has their percentage decreased? Similarly, the authors show that absolute numbers of CD4 and CD8 T cells change between sham and CLP mice, but how about their percentages among live splenocytes? Similarly, how about the percentage and absolute number of NK cells, and myeloid cells such as DCs, pDCs, macrophages and monocytes?

Response: As requested, corresponding percentages of CD4+ and CD8+ T lymphocytes among live splenocytes are now presented in Figure S2-A. While absolute counts were decreased, we did not observe any significant difference between sham and CLP mice in the percentages of CD4+ and CD8+ T cells among leucocytes although there was a tendency toward a decreased % of CD8+ T cells after CLP compared with sham.

In addition, proportions and absolute counts for naïve, central memory and effector CD4+ and CD8+ T lymphocytes were measured and are now presented in Figure S2 (Panels B to G). Among CD4+ T cells, absolute counts of naïve and effector T cells tended to decrease after CLP subpopulations while percentages of central memory cells increased in CLP mice. Among CD8+ T cells, absolute count of effector T cells significantly decreased after CLP.

Regulation of dendritic cells, NK cells, CD11b+ myeloid cells, mono/macrophages and neutrophils after CLP are now also presented in Figure S1. We observed a significant decrease of both percentages and absolute counts of dendritic cells and NK cells after CLP (Panels A and B in Figure S1). However myeloid cells, mono/macrophages and neutrophil counts were not significantly regulated in the spleen after CLP although a trend toward a higher proportion of neutrophils was observed (Panels C, D and E in Figure S1).

These results are also mentioned in the text (Page 6).

In Figure 4, do the mice in this figure also show differences in disease scores (as in Figure 1B) after treatment with Bortezomib?

Response: We did not observe any difference in disease severity score between mice treated or not with Bortezomib. Indeed, the small difference in weight loss observed between Bortezomib-treated or not mice after CLP might not translate in measurable differences in disease severity.

In Figure 6, the authors show the MFI of PD-1 and PD-L1 is increased in the PC of CLP mice. Does the percentage and absolute numbers of both PD-1+ and PD-L1+ PC show a similar trend?

Response: As requested, percentages of positive PD-1 and PD-L1 plasma cells are now shown in the Manuscript both for CLP mice (Fig 6D) and patients (Fig 7D-E). These results confirm the high PD-L1 expression on PCs after sepsis as both in mice and in patients.

Reviewer #2

Gossez et al., investigate the induction of regulatory plasma cells after sepsis and COVID-19, and their mechanism of T cell suppression (measured as proliferation of bulk, non-specified, T cell subsets). Using cecal ligation and puncture (CLP) model in mice they show that the CD1dhiCD5+ regulatory B10 cells are not affected post CLP, however the B220lowCD138+IgM+ regulatory Plasma cells (reg-PCs) accumulate after CLP mediated sepsis. Using ex-vivo T cell/reg-PC co-culture experiments they show that these reg-PCs suppress T cell proliferation in response to CD3/CD28 stimulation. Finally, they show that these reg-PCs use potentially both cytokine (IL-10 production) and cell-cell contact (PD-L1) mediated mechanisms to suppress T cell proliferation. They also found that the human bacterial sepsis and COVID-19 sepsis patients have increased representation of CD138+ reg-PCs that exhibit increased PD-L1 expression, and consequently suppress T cell proliferation ex-vivo.

A key gap in knowledge is addressed here – to mechanistically define cellular mechanism(s) that control sepsis-induced state of immunosuppression (or immunoparalyses). The conceptual strength is noted – to use mouse sepsis model to guide and inform limited exploration in human septic and/or COVID-19 (viral sepsis) patients to define and characterize the role of reg-PCs.

Overall, the study hinted at potentially interesting mechanism that could contribute to the sepsis-induced T cell dysfunction.

However, few major limitations are noted that significantly diminished enthusiasm of this manuscript in its current stage.

Response: We would like to thank this Reviewer for finding our results of interest. We provide important new information in the revised version of the Manuscript and in the point-by-point reply below to comments listed by this Referee.

Major Issues:

- The beauty of experimental mouse models is that variability in measures is relatively small enabling clear delineation and distinction between experimental groups/conditions... etc... It is puzzling where noted variability in a B6 mice comes from? Throughout the figures variability in control (sham) B6 mice is unexpectedly high – example Fig1D-F (ie. there is a mouse with 5×10^6 cells per spleen and mouse that has 20x more...). This should be explained and ramifications of that discussed with the data and conclusions drawn throughout.

Response: We would like to thank this Reviewer for this question regarding the variability of our results. Such apparently high variability can be explained by several aspects.

First, the surgical model of CLP was chosen to reproduce sepsis pathophysiology in mice. Indeed CLP is recognized as the most clinically relevant model of sepsis [Rittirsch *Nature Protocols* 2009]. However it is known to be more variable than other models of infection [Lewis *Surg Infect* 2016] because of the surgery procedure, the impact of ligation site and number and sizes of punctures. This variability can also be found in Sham animals as these were not naïve C57Bl/6 mice but were also subjected to anesthesia and surgery. Their caecum was exposed during the same amount of time as CLP mice before being replaced in the abdominal cavity. Finally sham animals received the same treatments (volume resuscitation, buprenorphine, antibiotics) as CLP mice. This control group of Sham animals allows to differentiate the impact of surgery versus infection on the immune system but they are not naïve mice. Second, variability can be attributed to our experimental design. In each experiment, batches of 3-5 animals were randomly allocated to experimental (CLP) and control (Sham) experimental groups and each experiment was reproduced 2-3 times independently. Results from all experiments were pooled and presented together. We did not apply any outlier exclusion process.

Third, variability can be attributed to technical / operator associated parameters such as improvement in technical skills between the initial experiments performed in this project and last experiments.

In total, we agree that our results do present some variability. We believe that this is not due to false or wrong experiments but rather to the complexity of our experimental model and design. This variability is observed in both experimental groups (Sham and CLP) and, despite this variability, we nevertheless observed differences between groups indicating that this does not preclude the conclusion that we obtained from this study.

- Sepsis-induced immunosuppression is long-lasting phenomenon that changes over time in septic survivors. Extending analyses post first few days after sepsis induction (weeks and months) should be considered.

Response: We would like to thank this Reviewer for this very interesting question. Indeed the persistence over time of sepsis-induced immune alterations has been previously shown in the literature even after ICU discharge of septic patients [Elek *Hum Immunol* 2023].

Unfortunately, the Animal Ethics Committee limited the follow-up of CLP mice in our experimental design to their percentage of weight loss. Indeed, a maximal weight loss of 15% of their initial weight was included in our animal CLP protocol as a human endpoint. As can be seen in Figure 1A, at D2, CLP mice have already lost more than 10% of the weight. Thus it is not possible to pursue the follow-up of these mice after this time point unless reducing the severity of the model.

In contrary, in patients included in the REALISM study, patient's follow-up extended till day 60 after ICU admission. In this cohort, we observed that although the peak of plasma cell expansion was at day 5, the percentage of circulating plasma cells was still significantly higher than in healthy donors at D14, D28 and D60 after ICU admission for those who were still in hospital. These data are presented in Figure 7A in the manuscript.

- T cell dysfunction is assessed here only through ability of T cells to proliferate in response to polyclonal stimulus. No attempts were made to define subsets of cells that are responding (and/or suppressed) – are they CD4 or CD8 T cells, are they naïve or effector or memory.... Many models are readily available to analyze the ability of Ag-experienced T cells to respond to their cognate antigens and provide effector functions - cytokine/chemokine production, killing capacity, protection to challenge/re-challenge afforded. These Measures are necessary to define T cell suppression.

Response: Again, we would like to thank this Referee for this question.

Complementary experiments were performed in an attempt to better characterize sepsis-induced T cell dysfunction.

First, proportions and absolute counts for naïve, central memory and effector CD4+ and CD8+ T lymphocytes were measured in CLP mice and sham animals and are now presented in Figure S2 (Panel B to G). Among CD4+ T cells, absolute count of naïve cells were decreased after CLP while the proportion of central memory increased. Among CD8+ T cells, effector T cells were decreased.

Second, the proliferation of CD4+ and CD8+ T cell subsets was evaluated in CLP mice and sham animals after *ex vivo* stimulation through TCR. We observed that both CD4+ and CD8+ proliferations were decreased after CLP (Fig S3A-B). In addition, in *ex vivo* co-culture experiments with plasma cells purified from patients, we showed that proliferations of both CD4+ and CD8+ T lymphocytes were simultaneously reduced when co-cultured with plasma cells from septic patients (Fig S5A-B).

Third, in addition to TCR-induced proliferation, we evaluated the proliferative response of splenocytes from CLP mice to a mitogen (i.e. phytohemagglutinin). We showed that mitogen-induced proliferation of T cells was also reduced in CLP mice. More specifically, the capacity of these cells to divide was lower after CLP compared with sham animals (Fig S3C).

Fourth, in addition to proliferation, T cell dysfunction was monitored through the release of IFN γ in supernatant after cell stimulation *ex vivo*. These novel analyses were performed not only after CLP in mice but also in patients in the model of *ex vivo* co-cultures. We showed that in mice (Fig 1I) and in patients (Fig 7I), decreased proliferation was accompanied by reduced IFN γ release in the supernatant.

Some of additional comments for authors to consider:

- The authors state that the regulatory PCs increase for up to 7 days in humans post bacterial sepsis and wanes after that. However, the kinetics of the reg-PCs and reg-B10 cells in mice is lacking and important in Fig.1. Providing this data can also help authors to justify the reason for selecting D2 for reg-PC characterization.

Response: Again we appreciate this question. However as mentioned previously, it is not possible to extend the follow-up period of mice in the current experimental design because of the weight loss and the ethical constraints related to our animal protocol. The monitoring of such aspect would require to

lower the severity of the CLP model that we used, which would most likely impact the results that we observed in the sense that PCs expansion might be less important.

- While the authors show in Fig.2 (A, B) that the reg-B10 cell %/# are not affected post CLP, they didn't test their T cell suppression potential. In Fig.2 F, the authors co-culture T cells with total B cells, however, the same experiment with the CLP reg-B10 cells will inform the readers about the contribution of both reg-B10 cells and reg-PCs.

Response: This Referee is correct that we did not evaluate the suppressive capacity of B10 cells in our model. Indeed, as in our model we did not detect any variation in the percentage of B10 lymphocytes after CLP or any suppressive activity of B cells isolated from CLP mice, we focused our efforts on the evaluation of plasma cells which percentages were significantly modified and presented with suppressive activity *ex vivo* after sepsis both in patients and in mice. The specific study of B10 cell subpopulation in sepsis can be the subject of dedicated studies.

*- In Fig 4, the authors conclude that the in vivo depletion of CD138+ reg-PCs improves T cell proliferation post CLP. While Bortezomib (BZB) is known to promote apoptosis in short- and long-lived PCs, this is still not a targeted depletion of CD138+ reg-PCs. Does Bortezomib influence re-B10 cell or other unknown Breg numbers and function, that can contribute to the observed result? Performing the sham and CLP surgery in *sdc-1*^{-/-} (CD138-ko) mice or administering anti-CD138 antibody in vivo one day post CLP could be a better experiment to strengthen the authors claims.*

Response: This Reviewer is correct that BZB is not a plasma cell specific therapy although it supposedly preferentially affect these cells because of their high protein synthesis activity. This is indeed because of its capacity to target plasma cells that bortezomib is used to treat multiple myeloma in clinic.

With that said, we indeed evaluated alternative methods to deplete plasma cells after CLP in mice. Since our goal was to deplete only PCs induced after CLP but not the entire pool of plasma cells present in mice; we did not evaluate *sdc-1*^{-/-} (CD138-ko) mice. However, we evaluated antibody depletion of plasma cells in some preliminary experiments. We tested the capacity of an anti-syndecan-1 antibody (clone 1A3H4, ref ab181789 from Abcam) administered IP at 4mg/kg at D1 after CLP to deplete sepsis-induced plasma cells. However, we did not observe any depletion. Thus we did not pursue these experiments and we focused on bortezomib injections.

- The improvement in weight loss observed in BZB treated CLP mice in comparison to DMSO treated CLP mouse is modest at best after 2 days post CLP (Fig 4.E). Collecting the weight loss and survival data for up to a week or longer with BZB could give a better significance to the authors claims.

Response: We agree that the difference in weight loss observed between BZB treated versus DMSO treated CLP mice is modest although significant and we appreciate the suggestion to extend the follow-up of these mice to amplify differences.

However, in this experiment set-up also, we were limited by ethical consideration and the 15% weight loss human endpoint. Thus, unfortunately, we were not able to perform suggested experiments.

Reviewer #1

The authors have successfully addressed most of the points raised in the first submission of this manuscript. However still a few new points, and some remain to be addressed:

Response: We would like to thank this Referee for acknowledging our thorough review efforts and his/her new suggestions.

Discussion points

New figure 8: Bortezomib can decrease CD4 T cells in bone marrow and spleen (doi: 10.1371/journal.pone.0135081). Hence, despite the claim that the Tregs are increased in plasma cell-depleted mice with bortezomib, it cannot be ruled out that this drug, a proteasome inhibitor, could have an effect on Treg effector function and not number, hence the observed results on T cell proliferation. The authors should acknowledge that the situation in vivo could be different despite the results in vitro shown in panels D, E and F. A Treg-deficient KO mouse model would definitively and undoubtedly answer this question.

Response: We would like to thank this Referee for this comment. We agree that, based on the experiments currently presented in the article, we cannot definitively rule out a direct effect of bortezomib on Treg functions. We fully agree that only Treg-deficient KO mouse model could answer this question. As this model is not yet available in our lab and could not easily and rapidly be performed within the context of this study, we were not able to perform these experiments within the scope of this paper. However, we acknowledge this limitation to our work and we now clearly mentioned this point in the Manuscript (Pages 14 and 20).

In addition, according to the author's new data, the presence of Tregs seems do not affect the proliferation of T cells. Indeed, comparison of 2nd of 4th column of panels D, E and F is as it seems not significant. How can the authors explain Tregs are not suppressive when they should be? Please include this in the discussion

Response: This referee is correct that the difference between T cell proliferation with or without Treg depletion was not statistically significant in Figure 8, Panels D, E and F. However,

we would like to point out that the calculated p values were close to significance (i.e. $p = 0.06$, non-parametric Wilcoxon paired test) and that we observed an increased proliferation in all pairs of samples except for one. This is illustrated in the Figure below in which proliferations of total T cells (left graph), CD4+ T cells (middle graph) and CD8+ T cells (right graph) were measured after stimulation of total T cells or T cells depleted of Treg. Results obtained in paired samples are linked by straight lines.

Thus we believe that this lack of significance may rather be due to a lack of statistical power than to a lack of effect. To better illustrate this aspect, we have now modified the figure format to present individual values as well as means and standard deviations (See Figure 8 R2). In addition, this aspect is also mentioned in the Discussion (Page 18).

Figure 6G and Figure S3: The results presented in these two figures have shared conditions (first and second column). However, they do not show the same levels (axes are also different). The results of these experiment should be carried out together and displayed together in a main figure.

Response: We would like to thank this Referee for this suggestion.

Corresponding experiments have now been carried out together and are presented on the same graph.

In this new set of experiments, we confirmed previously observed results. Indeed, addition of plasma cells purified from CLP mice to stimulated T cells purified from Sham animals was associated with a significant decrease of T cell proliferation when adding an isotypic IgG in

the culture medium ($p = 0.00020$, non-parametric Man Whiney test). In contrast, addition of an anti-PD-L1 blocking antibody at the same concentration was associated with a significant improvement of T cell proliferation compared with condition including isotypic IgG ($p = 0.0098$, non parametric Wilcoxon paired test). In line, Fig 6G has been updated and Fig S3 has been deleted.

Figure 7: Authors should use the same type of graph in panels H and I as in panel G. Like this the individual data points will be observable.

Response: Panels H and I of figure 7 have been modified to fit with data presentation in Panel G. Similarly, Fig S4 panels A and B have been modified to show individual and box-plots.

As nicely done for the mice in Figure 3, have the authors measure IFNg in the mice of figure 4?

Response: Unfortunately, we did not measure IFNg in the supernatant of splenocytes purified from mice treated or not with bortezomib and stimulated *ex vivo*. Such measurement would have been a confirmation of the results from proliferation experiment to show the restoration of normal T cell functioning in mice depleted of plasma cells after CLP.

Figure S2-B (right panel) is missing groups labels.

Response: Figure S2-B has now been modified. Thanks for indicating this typo.

A better readout than weight loss to assess acute sepsis response could be temperature and/or IL-1b, IL-6, TNF or CRP measurement in mouse. Have the authors considered these readouts?

Response: We fully agree that the monitoring of weight loss after CLP could be completed by other read-outs such as body temperature or plasma cytokine concentrations. Unfortunately, we did not include these read-outs in our models. They can certainly be considered in further experiments.

Line 228: authors mention Figure S3, but it should be figure S4.

Response: Again, we would like to thank you for pointing out this typo. In line with previous comment from this Referee, experiments corresponding to previous Fig S4 have now been carried out simultaneously with experiments presented in Fig 6G. Thus they are now presented together in the novel version of Fig 6G and Fig S4 has been deleted. Consequently, supplementary figures have been renumbered.

Reviewer #2

The authors provided detailed responses to major comments raised. However, one could argue that all 3 concerns were not adequately addressed.

Response: We would like to thank this Referee for acknowledging that we have answered to major concerns raised by Referees. We hope that novel experiments and additional modifications of the manuscript that we provide in this new version of the Manuscript will adequately address the Referee's remaining concerns.

1) Mouse to mouse variability is significant – much more than is published and available in the extensive literature on the subject. Sham mice are immunologically naïve, inbred, sex and age matched and anesthesia and/or surgery are not supposed to lead to those differences (20X differences in cellularity). Also, experiments performed here should not be considered as complex in design – they are rather simple and straightforward.

Response: Again, we apologize for this lack of clarity and we acknowledge the high variability in our results. We believe that such variability may come from the experimental design that was applied to this project. Indeed, we combined within the same figure results coming from different batches of experiments performed by different operators at different times without excluding any outlier. This is especially true for results presented in Fig 1; which recapitulate data obtained in a large number of mice over the entire duration of the study (n > 20 mice per group – pool of a maximum of 9 batches of experiments realized within a total of 5 years' period by 3 different operators: MG, CV, AV).

In total, we agree that our results present with high variability. This variability is observed in both sham and CLP groups. However, despite this high variability, we nevertheless observed significant differences between groups, which, in our opinion, supports the conclusions obtained from this study.

2) Despite limitations that they might have (15% weight loss) this manuscript is not defining sepsis induced immunosuppression using mouse CLP model.

Response: We agree with this comment and we would like to apologize for this overstatement. The term immunosuppression has thus been replaced by 'immune alterations' throughout the manuscript wherever referring to the CLP model (Pages 3, 5, 6, 7, 8, 16, 18 and 24).

3) Many models are readily available to analyze the ability of Ag-experienced T cells to respond to their cognate antigens and provide effector functions - cytokine/chemokine production, killing capacity, protection to challenge/re-challenge afforded. These measures are necessary to define T cell suppression. Those models are not utilized here - instead bulk, polyclonal stimulation is used which is limited in scope and reach.

Response: We would like to thank this Referee for this insightful comment. As suggested, we have now performed experiments evaluating the impact of CLP on Ag-experienced T cell functional response. We show that the *ex vivo* response (IFN γ production) to an antigen-specific stimulation of Ag-experienced T cells from CLP mice was significantly reduced compared with that of antigen-experienced T lymphocytes obtained from Sham animals. These results are now presented in Fig 1J and are mentioned in the Text (Page 7).

We would like to apologize if our previous answers to Reviewers' comments were not sufficiently detailed / satisfactory. Below, we provide a detailed point-to-point response to all remaining points raised by Reviewer #2. We also provide a revised version of the Manuscript in which changes are highlighted in red.

We have performed further analyses presented for Reviewers only to check and confirm the robustness of our data. We investigated IFN γ production at the single-cell level in CD4+ and CD8+ T cells after sepsis and we provided complementary measures of T cell fitness and immunity in CLP mice compared with Sham animals.

We sincerely hope that these modifications will be satisfying in regard with remaining comments from Reviewer #2.

Yours sincerely,

Fabienne Venet, on behalf of the other authors.

Reviewer #1

I think that despite not fully addressing the concerns, the authors have provided acceptable explanation and have discussed the limitation of this manuscript. I don't have any concerns.

Response: We would like to thank Reviewer #1 for finding our statement of limitations sufficient.

Reviewer #2

Point 1 was related to variability observed in their readouts using identical mice. Results shown are obtained by multiple investigators over many years of investigations - hence enormous variability. Are then the trends in data obtained by individual investigators the same?

Response: This question focuses on the variability of the results presented in the experimental mouse model of sepsis. Data presented in Fig 1D-F are cited as examples of variable results. In further analyses and as requested by this Referee, we have explored factors, which may have participated in this variability.

As mentioned previously, we chose to present in some panels of Figure 1 all the results of spleen cell immune monitoring available at the time of initial manuscript submission. This represented a pool of data coming from nine different experiments performed between 2017 (year 1) and 2020 (year 4) by three different experimenters for both surgery (AV or CV) and/or bench work (MG or CV).

In addition, we did not apply any outlier exclusion process. This experimental design thus combined 4 potential factors of variability: different batches of experiments (1) performed over a long period of time (2) by different surgery (3) and lab work (4) experimenters.

To evaluate the impact of each of these parameters on the variability of immune monitoring results, we performed principal component analysis of this set of data. Results are presented in Fig R1 below. Analyzed database combined measured values for total spleen cell count, percentages and absolute counts of CD4+ T lymphocytes, percentages and absolute counts of CD8+ T lymphocytes and percentages and absolute counts of CD4+ regulatory T lymphocytes obtained in 48 mice (n = 23 Sham and 25 CLP) during 9 different experiments by 2 surgery operators (AV, CV) and 2 bench work operators (MG, CV) over a period of 4 years (year 1 to year 4). This dataset strictly corresponds to the results presented in Fig 1D-E, Fig S1A, Fig S2A and Fig S2H.

Figure R1

Figure R1: Principal component analysis of immune monitoring data presented in Fig 1D-G and Fig S2A in Sham and CLP mice. Analyzed database comprised values of total spleen cell count, percentages and absolute counts of CD4+ T lymphocytes, percentages and absolute counts of CD8+ T lymphocytes and percentages and absolute counts of CD4+ regulatory T lymphocytes from 48 mice. Explicative parameters included (A) batch of experiment (expt#1 to expt#9), (B) year of experiment (year 1 = 2017, 2 = 2018 or 4 = 2020), (C) surgery operators (AV = Op1 or CV = Op 2) and (D) bench work operators (MG = Op1 or CV = Op2). Results are presented as individual values as well as mean point and 95% data ellipse for each group of explicative variable (minimum observations to plot an ellipsis = 4 – no ellipse is shown for expt#5). Principal components (PC1 – x-axis) and (PC2 – y-axis) are linear

combinations of the original variables that maximally explain the variance of all the dataset. PC1 and PC2 are shown, with 53.9% and 23% of variance explained, respectively. Coloring is set by group of explicative variable.

Figure R1A evaluated the effect of each experiment on the variance of this dataset. Mean values of each individual experiment clustered together at the center of the graph and respective ellipses were strongly overlapping. This suggests that none of 9 experiments significantly distinguished itself from the others. Figure R1B evaluated the effect of time, Figure R1C the effect of surgery experimenters and Figure R1D of lab work operators on results' variance. As in Fig R1A, clusters of data based on each of these parameters were overlapping. Only results obtained during year 4 by a different surgery and lab work operator (CV) tended to cluster together apart from the others thus potentially increasing variance of this dataset through inter-operator variability.

Next, we verified that, when normalizing results based on year of experiment (and thus of experimenters), differences between Sham and CLP groups were conserved. This analysis was performed for all the immune parameters presented in Fig 1D-E, Fig S1A, Fig S2A and Fig S2H. In addition, data obtained during reviewing process (i.e. year 5 by CV as surgery and lab work operator) but not presented in the paper were also included in this analysis. Representative results for CD4+ T cell count and percentages of regulatory T cells are presented in Fig R2 below.

Figure R2

Fig R2: Absolute count of CD4+ T lymphocytes and percentages of regulatory T cells in Sham and CLP mice based on years of experiments. Absolute counts of CD4+ T lymphocytes ($\times 10^7$ cells/ μ l) and percentages of CD4+CD25+CD127^{low} regulatory T cell (% among total CD4+ T lymphocytes) are presented. Red squares represent results in CLP mice and white circles in Sham animals. Geometric means with standard deviations (SD)

and individual values are shown. Results were separated based on the year of experiment performing from years 1 and 2 (experimenters: AV and MG) to years 4 and 5 (experimenter: CV). Results included in the black box were presented in Fig 1D-G. Results obtained during year 5 / reviewing process were not presented in Fig 1D-G.

Through this analysis, we confirmed that decreased absolute count of CD4+ T lymphocytes and increased percentages of Treg were systematically observed in CLP mice compared with Sham animals. Similar results were observed for total splenocyte and CD8+ T lymphocyte counts (lower in CLP mice) and for absolute counts of Treg (no difference between groups). Thus, through these additional analyses, we verified the consistency of our observations between Sham and CLP groups whatever the timing of experiment or the individual performing the surgery or lab work. Thus, this supports the robustness of the results presented in the article.

Last, it can be argued that, despite this normalization, a certain level of results' variability remains especially in the group of Sham animals. To further characterize this specific subset of data, we evaluated the distribution of this dataset. We applied Kolmogorov-Smirnov test to the set of data obtained in Sham animals during the 4 years of analysis (n = 36 mice). Results from this statistical analysis were not significant ($p > 0.05$) for all tested immune parameters showing that these set of data followed a normal / Gaussian distribution. In addition, we evaluated the presence of any outlier data for each immune parameter using 3 different statistical tests (Rout test, Grubbs test and iterative Grubbs test). None of these statistical analyses identified any outlier in this set of experimental data. This shows that, despite their variability, these data obtained in Sham animals are homogeneous. As mentioned before, although Sham animals are not submitted to cecal ligation nor puncture, they still follow identical surgical procedure including anesthesia and laparotomy. Cecum is isolated from abdomen and exposed during a period of time equivalent to CLP mice before being replaced in the peritoneal cavity and both layers of muscle and skin are sutured. It has been shown that manipulation of the bowel without puncture or ligation causes pro-inflammatory cytokines release and inflammation [PMID: 16213909]. In line, inflammatory changes with increased cytokine and chemokine levels have been described in Sham animals submitted to surgical trauma to the tissues [PMID: 21053304, - PMID: 27305321]. As prolonged surgical interventions with increased length of anesthesia as well as quality and consistency of the surgery procedure (size of the incision, trauma to the intestinal tissue) may lead to stronger inflammatory reactions in mice, these parameters may participate in the variability of results obtained in Sham animals. Indeed, as investigators become more familiar with the technique, the exteriorization of the cecum becomes easier and reduces the time of the procedure as well as possible damages to surrounding structures [PMID: 21053304, PMID: 33027848] . This suggests that the variability that we observed in the group of Sham animals is not unexpected especially considering the long period elapsed between first and last results presented in Fig 1. In addition, we also would like humbly to suggest that the variability observed in our Sham experiments is comparable to results

published in the literature by other groups of well-known investigators with extensive experience in the CLP model [for example see Fig 3 in PMID: 38754032].

Last, from a translational research perspective, we would like to suggest that such variability may even be considered as an advantage of this model. Indeed, extreme variability is noted in immune monitoring results obtained from septic patients [PMID: 31630991]. Results presented in Figures 7A-B in this article illustrated this high variability. Therefore, Rubio *et al* recently suggested that extreme standardization of animal models of sepsis needs to be balanced because excessive micromanagement creates an artificial, idiosyncratic environment that lowers multi-laboratory reproducibility [PMID: 31630991]. This was based on a study, which demonstrated that excessive within-study standardization is a major cause of poor reproducibility [PMID: 29470495]. More representative experimental design is required to improve the external validity and reproducibility of preclinical animal research and to prevent wasting animals and resources for inconclusive research. Suggestions included splitting experiments into multiple independent batches [PMID: 29505576]. The authors concluded that the more variation was introduced, the more reliable and translatable the finding would be.

In total, we hope that these complementary analyses convincingly illustrate the robustness of the results presented in this manuscript. However, we acknowledge that our choice of experimental design to present pooled data coming from multiple experiments performed over a lengthy period by different experimenters surely has increased this variability. We are convinced that, because significant differences between Sham and CLP groups could still be observed, this reinforces their robustness and the clinical relevance of these results. However, to insure full transparency regarding these different aspects, this limitation of our work is now clearly mentioned in the text. Material and Methods (Page 22) and Discussion / Limitation sections (Pages 20-21) of the Manuscript have been modified. Also, if requested by the Editor, Figures R1 and R2 presented above could be included in the Supplemental Material or Source data documents.

Point 2: Study of long effect of CLP induced sepsis

Point 2 was related to the fact that they do not study long(er)effects of CLP induced sepsis (immunosuppression). Instead of extended their experiments to include later time points the authors decided to replace term 'immunosuppression' with 'immune alterations'. This manuscript then do not address immunosuppression.

Response: This question from Reviewer 2 relates to the use of the term ‘Immunosuppression’ in our manuscript in regard with the duration of the immune monitoring after sepsis.

In patients, sepsis-induced immunosuppression is defined by the presence of any immune alterations (i.e. modification of any phenotypic or functional aspect of immune response compared with normal values) in association with deleterious outcomes [PMID: 24232462, PMID: 38142698, PMID: 29225343]. In the current study, we showed that patients with either bacterial or viral sepsis presented with well-described sepsis-induced immune alterations such as decreased expression of monocyte HLA-DR and CD4+ T cell lymphopenia at D3 after sepsis (See Table S1) and the novel description of an increased percentage of circulating plasma cells with regulatory function (Fig 7A-B-C). Regarding this latter immune alteration, we demonstrated that increase in circulating plasma cells persisted in patients at least 2 months after the onset of shock (See Fig 7A) and was associated with increased mortality (See Fig 7B). Thus we believe that these data efficiently describe the development of sepsis-induced immunosuppression in these cohorts.

In mice, we showed that the murine model of sepsis induced by CLP as set up during this study efficiently recapitulated these immune alterations induced by sepsis in patients. This was the objective of data presented in Figure 1. Unfortunately, the Animal Ethics Committee of our institution required the humane endpoint of a maximal weight loss of 15% after CLP to be included in the animal protocol as a matter of animal well-being. Although mortality is very low in this CLP model, CLP mice usually lose more than 10% of their initial weight at D2 (Fig 1A) and more than 15% at D3 after CLP and thus have to be sacrificed. In this context, to maintain CLP mice alive for weeks and months after surgery, Animal Ethics Committee from our Institution requires major modifications of the current animal protocol with a reduction of CLP severity (smaller size of the needle, smaller length of the ligation) and the removal of cecal ligation during a second intervention (to avoid the formation of any abscesses around the ligated cecum). We anticipate that the implementation of such important modifications of the CLP protocol would affect the clinical relevance of the model and the extent to which it recapitulates immune alterations observed in patients. However, we acknowledge that, because of the lack of these long-term follow-up data in mice, the term immunosuppression could not be strictly applied to this murine model of sepsis. Thus, in the current version of the Manuscript, we only refer to the CLP model as a model of sepsis, which recapitulated early immune alterations observed in patients.

This has been modified throughout the Manuscript (Abstract: Page 3, Introduction: Page 5, Results: Pages 6-7-8, Discussion: Pages 15-16-18, Conclusion: Pages 20-21, Material and Methods: Page 27) and this is listed as a limitation of this model in the Discussion part of the Manuscript (Pages 20-21).

Point 3: Breadth of T cell analyses in the CLP model

Point 3 was related to breadth of T cell analyses. One panel of a new data is provided. Resolution afforded with the system used is minimal and does not address per cell capacity to provide cytokines in response to cognate Ag-stimulation. Other measures of T cell fitness and immunity are not provided. Many mouse models of infection exist that provide plenty of pathogen-specific T cells to study and none was used here.

Response: This remaining point raised by Reviewer 2 relates to the extent of T cell dysfunctions characterization in the CLP mice model.

Following initial recommendations from Reviewer #2, a number of additional experiments were performed to better characterize T cell alterations induced by CLP in complement with the results initially presented in the Manuscript. First, the phenotype of lymphocyte subsets present in the spleen after CLP was improved as we monitored naïve, central memory and effector CD4⁺ and CD8⁺ T lymphocytes in CLP mice and sham animals. We showed that among CD4⁺ T cells, absolute count of naïve cells were decreased after CLP while the proportion of central memory increased. Among CD8⁺ T cells, effector T cells were decreased (See Fig S2). Second, we improved the characterization of T lymphocyte functional alterations not only in the model of CLP but also in patients after *ex vivo* co-culture of T lymphocytes with purified plasma cells. We showed that the proliferation rates of both CD4⁺ and CD8⁺ T lymphocytes were decreased not only after CLP but also after *ex vivo* co-culture experiments with plasma cells purified from patients (See Fig S3 A-B and Fig s4). Third, in addition to TCR-induced proliferation, we evaluated the proliferative response of splenocytes from CLP mice to a mitogenic stimulation (i.e. phytohemagglutinin). We showed that mitogen-induced proliferation of T cells was also reduced in CLP mice. More specifically, the capacity of these cells to divide was lower after CLP compared with sham animals (See Fig S3C). Fourth, in addition to T cell proliferation, we added another read-out of T cell dysfunction through the monitoring of IFN γ release in supernatant after cell stimulation *ex vivo*. These novel analyses were performed not only after CLP in mice (See Fig 1I) but also in the model of *ex vivo* co-cultures in mice (Fig 2H and Fig 3K) and in patients (See Fig 7I). We showed that in mice and in patients, decreased proliferation was accompanied by reduced IFN γ release in the supernatant. To note, in these experiments, we observed that IFN γ release in supernatant after cell stimulation *ex vivo* was strongly correlated to individual cell proliferation (either total T lymphocytes or CD4⁺ and CD8⁺ T cell subsets). This was observed not only in co-culture experiments with purified T cells and B or plasma cells but also in experiments with total splenocytes. Results from correlation analyses are shown in the Table R3 below.

Correlation of IFN γ concentration in supernatant vs	% of proliferating T lymphocytes	% of proliferating CD4+ T lymphocytes	% of proliferating CD8+ T lymphocytes	Corresponding experiments
Spearman R	0.7174	0.7319	0.7484	Splenocyte stimulation experiments in CLP and Sham animals (Fig 1I and Fig S3C)
p-value	< 0.0001	< 0.0001	< 0.0001	
Number of values	30	30	30	
Spearman R	0.86	0.8651	0.8513	Co-culture experiments in septic patients (Fig 7H-I)
p-value	< 0.0001	< 0.0001	< 0.0001	
Number of values	38	38	38	
Spearman R	0.8875			Co-culture experiments in CLP and Sham animals (Fig 2H – Fig 3K)
p-value	< 0.0001			
Number of values	44			

Table R3. Spearman correlations between IFN γ concentration in cell culture supernatants and T lymphocyte proliferation. IFN γ concentration was measured by ELISA in cell culture supernatants and results were expressed as pg/mL. CD3+ and/or CD4+ and/or CD8+ T lymphocyte proliferation was measured by flow cytometry either through CellTrace dilution (mice experiments) or EdU incorporation (human experiments) and results were expressed as percentages of proliferating cells among total respective population. Experimental design (i.e. stimulation and co-culture conditions) are presented in corresponding figures in the Manuscript. Non parametric Spearman correlation test was applied and corresponding R and p-values are shown.

Fifth, we examined the impact of CLP on antigen-specific T lymphocyte response by monitoring IFN γ release after *ex vivo* re-stimulation with OVA antigen of splenocytes isolated from OVA-immunized mice. We showed that IFN γ release induced by this OVA peptide antigenic re-stimulation *ex vivo* was decreased in splenocytes isolated from CLP OVA-immunized mice compared with Sham OVA-immunized animals (See Fig 1J).

To complement these data and to provide other measures of T cell fitness and immunity in this model, we now have additionally monitored the proportions of apoptosis and necrosis among T cells in the spleen of mice after CLP. As previously described, we showed that the percentages of apoptotic and necrotic cells among CD4+ and CD8+ T splenocytes were higher in CLP mice than in Sham animals [PMID: 9110409, PMID: 10447713]. This was concordant with the significant CD4+ and CD8+ T lymphopenia observed in the spleen of mice after CLP. This is now presented in the Results part of the Manuscript (Page 6) and in Figure 1F.

In addition we explored the expressions on remaining T lymphocytes in the spleen of activation (CD69) / exhaustion (PD-1) markers. We observed that the expressions of these markers were significantly increased on CD4+ and CD8+ T lymphocytes from CLP mice compared with Sham animals. This is concordant with previously published results [PMID: 32890312 - PMID: 20483923 - PMID: 31306346]. Regarding increased PD-1 expression, as this was associated with reduced functionality (decreased proliferation in response to re-stimulation *ex vivo*), this describes the presence of CD4+ and

CD8+ T lymphocytes exhaustion in spleen of mice after CLP as described by Pr R Hotchkiss [PMID: 23663657]. This is now presented in the Results part of the Manuscript (Pages 6-7) and in Figure 1G.

Finally, in the experiments exploring the impact of CLP on antigen-specific response in OVA-immunized mice, in addition with the measurement of IFN γ concentration in stimulated splenocyte supernatants, we have now explored at the single cell level the capacity of CD4+ and CD8+ T splenocytes to produce IFN γ by performing intracellular staining by flow cytometry. With these new experiments, we confirmed that, upon re-stimulation *ex vivo*, the numbers of IFN γ producing CD4+ and CD8+ T splenocytes were significantly reduced after CLP compared with Sham animals. A similar decrease was observed regarding the percentages of IFN γ secreting CD4+ and CD8+ T splenocytes. This is now presented in the Results part of the Manuscript (Page 7) and in Figures 1K and S3D.

Thus, we hope that these additional parameters describing T cell response have helped to largely improve the characterization of T cell alterations induced by sepsis in the CLP model.

Last, we fully agree that we did not explore the pathogen-specific T cell response in this model. Indeed, as murine models of pathogen-specific infections are not currently running in our lab, their set-up including the validation of the animal protocol by Animal Ethics Committee would have required an extensive period of time. In addition, we would like to underline that the initial objective of this murine CLP model was to be clinically relevant and thus to recapitulate immune dysfunctions observed in septic patients and that such evaluation of the impact of sepsis on antigen-specific T cell response is never assessed in patients.